This is a Registered Report and may have an associated publication; please check the article page on the journal site for any related articles.

# Association of hyperuricemia with coronary heart disease and other cardiovascular outcomes: A systematic review and dose-response meta-analysis

Diyang Lyu[1☉], Rui Zhuang[1☉], Jiaqi Li[1☉], Yucen Wu[2], Yiming Di[3], Meifen Song[3], Liyong Ma[1], Jingen Li[1*], Yong Zhang[4,5*]

1 Department of Cardiovascular Medicine, Dongzhimen Hospital, Beijing University of Chinese Medicine, Beijing, China, 2 Department of Rehabilitation, Dongzhimen Hospital, Beijing University of Chinese Medicine, Beijing, China, 3 College of Basic Medical Sciences, Shanxi University of Chinese Medicine, Shanxi, China, 4 Dongzhimen Hospital, Beijing University of Chinese Medicine, Beijing, China, 5 The seventh affiliated hospital of Xinjiang Medical University, Xinjiang, China

☉ These authors contributed equally to this work.
* lijingen198908@126.com (JL); zhangyong_tcm@163.com (YZ)

## Abstract

### Introduction

Uric acid (UA) is considered as a potential risk factor for coronary heart disease (CHD) and other cardiovascular diseases (CVDs). However, the association between hyperuricemia and the risk of CHD and other cardiovascular outcomes has not been fully clarified. This systematic review and dose-response meta-analysis was conducted to comprehensively the association between hyperuricemia and the risk of CHD or other cardiovascular outcomes in the general population.

### Methods

We systematically searched Medline, Cochrane Library, Embase, and two clinical trial registration databases from inception to June 30, 2025, without restrictions on language or publication status. Only cohort and case-control studies enrolling participants without CHD, other CVDs, or gout at baseline were included. The primary outcome was the association between hyperuricemia and the risk of CHD, and secondary outcomes were the association between hyperuricemia and the risk of fatal and nonfatal CVDs, included CHD death, CVD, CVD death, and myocardial infarction (MI). Risk of bias was assessed using the Risk Of Bias In Non-randomized Studies-of-Exposure (ROBINS-E) tool. All statistical analyses were performed using R 4.4.2. We conducted meta-analyses, heterogeneity assessments, publication bias tests, trim-and-fill analyses, subgroup and sensitivity analyses, meta-regressions, and dose-response meta-analyses. The GRADE recommendation was used to evaluate the quality of evidence.

**Data availability statement:** All relevant data are contained within the paper and its Supporting Information files.

**Funding:** This work is sponsored by China Postdoctoral Science Foundation under Grant Number 2024M750266; the Postdoctoral Fellowship Program of CPSF under Grant Number GZC20240164; Beijing Nova Program (20240484736); and Beijing University of Chinese Medicine affiliated Dongzhimen Hospital's Youth Backbone Talent Program (DZMG-QNGG007). Funders of this study have no role in the study design, the collection and analysis of the data or the writing of the manuscript.

**Competing interests:** The authors have no conflicts of interest to disclose.

## Results

A total of 42 articles representing 39 individual studies and 1,082,880 participants were included. Among these, 2 articles were assess as "very high risk of bias", eight as "high risk of bias", and two as "some concerns". Hyperuricemia was significantly associated with an increased risk of CHD [HR 1.21 (95%CI 1.14–1.28), $p < 0.001$, I2 = 34.34%], CHD death [1.20 (1.05–1.36), $p = 0.005$, I2 = 41.28%], CVD death [1.75 (1.12–2.74), $p = 0.014$, I2 = 49.48%], and MI [1.23 (1.03–1.47), $p = 0.025$, I2 = 56.96%]. No significant association was observed for overall CVD risk [1.09 (0.94–1.27), $p = 0.245$, I2 = 0%]. For each unit increase in serum UA, the risk of CHD, CHD death, CVD, CVD death, and MI increased by 16%, 13%, 12%, 11%, and 7%, respectively. No factors with a significant impact on the results were identified through subgroup analyses or meta-regression. Sex may have a potential influence, but the results were not robust. Further dose-response meta-analysis revealed a linear relationship between higher serum UA and CVD risk, and a U-shaped association between serum UA and CVD mortality in men. The quality of evidence was rated as low for CHD and very low for the other cardiovascular outcomes.

## Conclusion

This systematic review and dose-response meta-analysis provides low- to very-low-quality evidence suggesting that hyperuricemia may be associated with an increased risk of CHD and other fatal or nonfatal CVDs.

## Trial registration

This study was registered in PROSPERO CRD42024538553.

## Introduction

Coronary heart disease (CHD) is a major component of cardiovascular disease (CVD) and represents the leading cause of morbidity and mortality worldwide, contributing substantially to the global health burden [1]. The pathophysiology of CHD is multifactorial, encompassing a complex interplay of traditional modifiable and unmodifiable risk factors including age, sex, hypertension, hyperlipidemia, smoking, and diabetes, etc. [2] In recent years, metabolic factors have been increasingly recognized as modifiable risk factors for CVD based on recent clinical studies [3]. Uric acid (UA), traditionally considered an inert metabolic end-product of purine metabolism, has been shown to associated with several chronic diseases, especially the cardiovascular and renal diseases [4,5].

Historically, Elevated serum UA levels, a condition known as hyperuricemia, have been primarily associated with gout, a type of inflammatory arthritis caused by the deposition of monosodium urate crystals in joints [6]. Patients with gout, regardless of CHD or other CVD, typically receive urate-lowering therapy. Gout is characterized by significant inflammation, which differs from the majority of patients with asymptomatic

hyperuricemia [6]. This distinction raises uncertainty regarding whether early intervention for asymptomatic hyperuricemia is necessary for cardiovascular benefit [7]. Therefore, clarifying the association between asymptomatic hyperuricemia and the occurrence of CHD or CVD is important.

Hyperuricemia has been implicated in various cardiovascular disorders, including CHD, but the precise nature of this relationship remains contentious within the existing literature [8–10]. Differences in study design [11–13], sample sizes [14,15], or sex [16] often explain variability in findings. For example, some studies have reported a significant association between elevated serum UA and an increased risk of CHD or CVD [15], whereas others have not detect such an association [17], leading to conflicting clinical recommendations. Confounding factors, such as lifestyle, medications, and comorbidities, further complicate interpretation [18].

Recent high-quality studies [19–22] with large sample sizes, long follow-up periods, and robust methodologies offer an opportunity to reassess the association between UA levels and CHD or CVD onset. These studies report additional outcomes, including fatal and nonfatal CHD, CVD, and myocardial infarction (MI), in general or single-sex population [19,23,24]. It is critical to examine not only the association between hyperuricemia and CHD development, but also its potential roles in the incidence and mortality of broader CVD and MI outcomes.

This systematic review and dose-response meta-analysis aims to comprehensively synthesize evidence on the association between hyperuricemia and fatal and nonfatal CHD as well as broader cardiovascular outcomes, including CVD and MI incidence and mortality. By focusing on cohort and case-control studies including populations without prior CHD or gout, this study aims to clarify the impact of UA levels on cardiovascular outcomes. Moreover, by addressing inconsistencies in previous studies, the current analysis intends to knowledge gaps and provide evidence-based recommendations.

## Methods

This systematic review and dose-response meta-analysis was performed according to a published protocol [25] and following the guidance of the Cochrane Handbook [26]. We reported our study according to the latest Preferred Reporting Items for Systematic Reviews and Meta-Analyses (PRISMA) statement [27] and the Meta-analysis Of Observational Studies in Epidemiology (MOOSE) guidelines [28]. This study has been registered on Prospective Register of Systematic Reviews (PROSPERO) (CRD42024538553).

### Search strategy and eligible criteria

Two reviewers (DL and RZ) independently searched the electronic databases including Medline, Cochrane Central Registry of Controlled Trials (CENTRAL), and Cochrane Databases of Systematic Reviews through Ovid, and Embase, Registers ClinicalTrials.gov and International Clinical Trials Registry Platform (ICTRP) through each website. The registries were also searched for unpublished studies or missing data. We searched from inception to June 30, 2025 with the established search strategy described in our protocol [25]. Reference lists of relevant articles were screened manually. Records in other languages were fully reviewed with Google Translate. No automated tools were used at this step. Any disagreement was solved by a third reviewer (JinL) and further discussed among all reviewers if unresolved.

Retrieved studies were screened independently by the two reviewers according to the following inclusion and exclusion criteria described previously [25]:

1. Participants: We included studies that enrolled human adults without prior CVD or gout. The study designs of the original studies varied; thus, we did not set limitation on the diagnose of hyperuricemia. We also set no limitation on the sex, human race, region or country, etc. However, original studies that enrolled only participants with hyperuricemia were excluded.

2. Exposure: The exposure is the occurrence of hyperuricemia or different UA levels among the participants during the follow-up period. The included studies should report the exact value or range of serum UA and the assessment method

to test the serum UA. If the exposure was hyperuricemia, the original study should provide the cut-off value of serum UA. Studies involved urate-lower therapies were excluded.

3. Control: The participants regarded as the control group are those never diagnosed with hyperuricemia or with relative low level of serum UA in the study.

4. Outcome: The outcomes included the hazard ratio (HR), odds ratio (OR), or risk ratio (RR) of fatal or nonfatal CHD, CVD, or MI to evaluate whether hyperuricemia or higher serum UA level is a risk factor.

5. Study design: We only included retrospective or prospective cohort human studies or case-control human studies. The included studies should have at least 100 participants of sample size, with at least 1 year of follow-up period.

### Definition of outcomes

In this study, we investigated the association between hyperuricemia or elevated serum UA level and fatal or nonfatal CHD, CVD, and MI. The outcome indicators included HR, OR, or RR. The primary outcome was the association between hyperuricemia and CHD onset. Since several studies investigated the association between serum UA and CVD (defined CVD as a series of diseases including CHD, MI, stroke, etc.) or MI, we also included them as secondary outcomes, although these secondary outcomes were not mentioned in our study protocol. Studies reported outcomes stratified by sex or per unit increase in serum UA level were also included.

### Data extraction and quality assessment

Two independent reviewers (DL and JiaL) went through the included original studies to extract information and data. The following information was extracted as our protocol described [25]: publication information, demographic information, study design information, methodological information including the statistical models and the confounding factors involved in the statistical models, and other information. We also extracted original data including the HR/OR/RR with 95%CI, case numbers, and group sizes. We further contacted the corresponding authors or the first authors for missing information when the e-mail address was available. We failed to obtain any individual patient-level data to perform individual patient data meta-analysis.

We assessed the risk of bias of the included observational studies using the Risk Of Bias in Non-randomized Studies – of Exposure (ROBINS-E) tool [29]. The overall risk of bias of each original study was determined by the seven domains of risk of bias: risk of bias due to confounding, risk of bias arising from measurement of the exposure, risk of bias in selection of participants into the study (or into the analysis), risk of bias due to post-exposure interventions, risk of bias due to missing data, risk of bias arising from measurement of the outcome, risk of bias in selection of the reported result. Two reviewers (DL and YZ) assessed all of the included original studies. Since the R package robvis did not support the ROBINS-E tool, and the online version of robvis did not include the judgement "Low Risk of Bias, except for concerns about residual confounding", we finally generated the traffic light plot and the summary plot to present the quality assessment with our own Python script.

### Data analysis

Quantitative analyses were performed for the outcomes. Firstly, we divided the data into 3 parts: pooled estimation of hyperuricemia, the pooled estimation of per unit increase of serum UA, and dose-response meta-analysis. For studies reported tertile, quartile, or quintile of serum UA levels, data from the highest UA group or those with serum UA≥ 7 mg/dL were used in the pooled estimation of hyperuricemia, and further excluded in sensitivity analysis. Pooled estimation of per unit increase of serum UA included all results regardless of the exact range of the unit, and further excluded those non-1 mg/dL data for sensitivity analysis. Due to heterogeneity in study design and demographics, we used the random

effects model for all pooled estimations regardless of the assessed heterogeneity to achieve more reliable results. The $I^2$ and $\tau^2$ statistics were used to assess the heterogeneity, while Peter's test (n ≥ 10), Begg's test (n ≥ 3), and Egger's test (n ≥ 3) were employed to test the potential publication bias. We also conducted the trim-and-fill [30] analysis to impute "missing studies" and assess potential publication bias as an additional approach for each pooled estimation, and presented with funnel plot. Statistical analyses were performed using R 4.4.2 software (http://www.r-project.org) with the package "meta" (v8.1-0).

### Dose-response meta-analysis

We performed the dose-response meta-analysis with the R package "dosresmeta" [31] (v2.2.0). We extracted quantile-based data from the included studies. For each original study, the sample size of each group was multiplied by the average follow-up years to obtain the amount of person-time, and the level of serum UA for each group was determined by calculating the average of corresponding upper and lower bounds of serum UA quantile. For single-bound quantiles, 1 mg/dL was added or subtracted. When establishing the dose-response model, we first established a linear model; for nonlinear models, since this review included a study with tertile data [12], we established a restricted cubic spline (RCS) model with 3 knots in our study. We performed a nonlinear trend test on the RCS model: if the relationship between serum UA level and the specific cardiovascular outcome is linear, the result of the linear model would be reported, otherwise the result of the RCS model would be reported.

### Subgroup analysis

We further performed subgroup analysis for both traditional meta-analysis and dose-response meta-analysis. Previous systematic review reported sex difference in the association between serum UA and cardiovascular outcomes [16]. Thus, we extracted data of male or female from original studies, and calculated the pooled estimations of fatal or nonfatal outcomes for specific sex. In addition, we found that there was only 1 article that reported their finding in OR, 6 reported in RR, 4 did not report, while the other 31 reported in HR (Supplementary S1 Tabele in S1 Appendix). Therefore, we also performed subgroup analysis by RR or HR, to further explore the influence of different statistics. The heterogeneity, potential publication bias, and trim-and-fill test were also conducted for each subgroup when available.

### Meta-regression

We conducted meta-regression to identify potential factors that might be associated with the risk of cardiovascular outcomes. This was not prespecified in our study protocol [25], representing a deviation from the original plan. Given the incomplete reporting of original data in the included studies (Table 1 and Table S1 in S1 Appendix) and the relatively small number of studies contributing to each meta-analysis, only age, sex, and body-mass index (BMI) were included in the meta-regression. For the analysis based on per unit increase of serum UA, data availability allowed inclusion of only age and sex; therefore, we performed additional meta-regression using these two variables in the meta-regression of hyperuricemia as a sensitivity analysis. Considering the high proportion of missing data, no imputation was performed; instead, unavailable data were directly excluded from the analysis.

### Assessment of the quality of the evidence

We assessed the quality of each outcome's evidence quality following the guidance of the Grading of Recommendations Assessment, Development and Evaluation (GRADE) recommendation [32]. Since this systematic review included only cohort studies and case-control studies, the initial quality of evidence was "Low" for each outcome, and subsequently we used the following items when appropriate to downgrade the quality of evidence: risk of bias, inconsistency, indirectness, imprecision, and publication bias. On the other hand, the following items were used to upgrade the quality of

**Table 1. Main characteristics of the included studies.**

| Study | Participant source | Age (mean, SD) | Sex (male) (%) | Sample size | Population feature | Follow-up period (years) | Serum uric acid test time point | Serum uric acid assessment method | Outcome | Number of nonfatal outcomes | Number of fatal outcomes | Statistical model (number of adjusted factors) | Study design |
|---|---|---|---|---|---|---|---|---|---|---|---|---|---|
| Fessel 1980 | American | 37.2, 7.05 | 55.70% | 304 | Population without diabetes, kidney disease, rheumatoid arthritis, or other conditions known to be associated with hyperuricemia. | 9 | Baseline | Phospho-tungstic acid method | CHD | 7 | | Chi-square analysis | Prospective cohort study |
| Goldberg 1995 | Honolulu Japanese | Not mentioned. | 2710 (100%) | 2710 | General population. | 20 | Baseline | Phospho-tungstic acid method | CHD | 352 | | Cox | Prospective cohort study |
| Wannamethee 1997 | British | Not mentioned. | 5757 (100%) | 5757 | General population. | 18 | Baseline | Colorimetric method | CHD | 518 | | Cox | Prospective cohort study |
| Culleton 1999 | Framingham residents of America | 47.09, 15.58 | 3075 (45.47%) | 6763 | General population. | 20 | Not mentioned. | Phospho-tungstic acid method | CHD and CVD death. | 617 | 429 | Cox | Prospective cohort study |
| Liese 1999 | Germany | 54.10, 5.84 | 1074 (100%) | 1074 | General population. | 8 | Baseline | Colorimetric method | CVD death and MI. | 60 MIs. | 44 | Cox | Prospective cohort study |
| Fang 2000 | Non-Hispanic American | 48.1, 14.0 | 2702 (45.60%) | 5926 | General population. | 16.4 (average) | Baseline | Phospho-tungstic acid method | Death for cardio-vascular disease and ischemic heart disease. | | 731 | Cox | Prospective cohort study |
| Moriarity 2000 | American communities | Not mentioned. | 5904 (43.7%) | 13504 | General population. | 8 | Baseline | Uricase method | CHD | 392 | | Cox | Prospective cohort study |
| Jee 2004 | Korean | 44.6, 8.7 | 22698 (100%) | 22698 | General population. | 9 | Baseline | Not mentioned. | ASCVD death | | 323 ASCVD death (99 IHD and 192 stroke) | Cox | Prospective cohort study |
| Baibas 2005 | Greek | Not mentioned. | 504 (43.83%) | 1150 | General population | 15 | Baseline | Colorimetric method | CHD death | | 67 | Cox | Prospective cohort study |
| Chien 2005 | Chinese | Not mentioned. | 1673 (47.09%) | 3602 | Including participants with CHD, then excluded in corresponding data analysis. | 11 | Baseline, first (2 years) and fifth (7 years) follow-up. | Colorimetric method | CHD | 86 | | Cox | Prospective cohort study |

*(Continued)*

| Study | Partic-ipant source | Age (mean, SD) | Sex (male) (%) | Sample size | Population feature | Follow-up period (years) | Serum uric acid test time point | Serum uric acid assess-ment method | Outcome | Num-ber of nonfa-tal out-comes | Number of fatal out-comes | Statis-tical model (num-ber of adjusted factors) | Study design |
|---|---|---|---|---|---|---|---|---|---|---|---|---|---|
| Bos 2006 | Nether-lands, Rotter-dam | 69 | 1552 (35.4%) | 4385 | General population. | 8.4 (average) | Baseline | Uricase-perox-idase method | CHD | 515 (194 MI) | | Cox | Prospec-tive cohort study |
| Gerber 2006 | Israeli male employ-ees | 49, 7 | 9125 (100%) | 9125 | General population. | 23 | Baseline and after 5 years. | Phospho-tungstic acid method | CHD death | | 830 | Cox | Prospec-tive cohort study |
| Iwashima 2006 | Japanese | 61.40, 1.35 | 296 (47.82%) | 619 | Hypertensive participants. | 2.8 (average) | Baseline | Uricase-perox-idase method | CVD events. | 28 | | Cox | Prospec-tive cohort study |
| Baba 2007 | Japanese | 62.70, 9.06 | 810 (40.02%) | 2024 | Atomic bomb survivors. | 10 | Baseline | Uricase-perox-idase method | CHD | 49 | | Cox | Prospec-tive cohort study |
| Strasak 2008a | Austrian | 41.6, 14.7 | 83683 (100%) | 83683 | General population. | 21 | Baseline | Uricase method | CHD death | | 1699 | Cox | Prospec-tive cohort study |
| Strasak 2008b | Austrian | 62.3, 8.8 | 0 (0%) | 28613 | General population. | 21 | Baseline | Uricase method | CVD death and CHD death | | CVD death 2874 | Cox | Prospec-tive cohort study |
| Chen 2009 | Chinese | 51.54, 11.48 | 41879 (46.33%) | 90393 | General population. | 9 | Baseline | Uricase-perox-idase method | CVD death and CHD death | | CVD death 1151, CHD death 286 | Cox | Prospec-tive cohort study |
| Holme 2009 | Swedish | 48.15, 11.76 | 221178 (52.95%) | 417734 | General population. | 17 | Baseline | Uricase method | AMI | 17174 | | Cox | Prospec-tive cohort study |
| Chuang 2012 | Chinese | 42.38, 14.13 | 59960 (46.64%) | 128569 | General population | 7.33 (average) | Baseline | Uricase-perox-idase method | IHD | 2049 | | Cox | Prospec-tive cohort study |
| Kawai 2012 | Japanese | 61.9, 0.5 | 369 (55.16%) | 669 | General population | 7.1 (average) | Baseline | Not men-tioned. | CVD | 58 | | Cox | Prospec-tive cohort study |
| Kivity 2013 | Israeli | 50.56, 9.18 | 6580 (72.00%) | 9139 | Nondiabetes population. | 4.8 (average) | Baseline | Not men-tioned. | CVD | 889 | | Cox | Retro-spective cohort study. |
| Onat 2013 | Turkish commu-nity | 51.69, 10.50 | 653 (47.63%) | 1371 | Nondiabetes population. | 4.9 (average) | Baseline | Uricase-perox-idase method | CHD | 136 | | Cox | Prospec-tive cohort study |

*(Continued)*

| Study | Participant source | Age (mean, SD) | Sex (male) (%) | Sample size | Population feature | Follow-up period (years) | Serum uric acid test time point | Serum uric acid assessment method | Outcome | Number of nonfatal outcomes | Number of fatal outcomes | Statistical model (number of adjusted factors) | Study design |
|---|---|---|---|---|---|---|---|---|---|---|---|---|---|
| Shiozaki 2013 | Japanese police officers | 46.77, 5.25 | 174 (100%) | 174 | General population | 5 | 5 years earlier. | Not mentioned. | CHD | 58 | 2 | Multivariate logistic regression | Retrospective case-control study. |
| Storhaug 2013 | Norwegian residents | 59.68, 10.25 | 2696 (47.30%) | 5700 | General population | 15 | Baseline | Uricase-peroxidase method | MI | 659 | | Cox | Prospective cohort study |
| Puddu 2014 | Italian resident | 53.72, 0.65 | 1273 (44.08%) | 2888 | General population | 13.5-19.5 | Baseline | Colorimetric method | CVD, CHD, CVD death and CHD death | CVD 383, CHD 189 | CVD death 321, CHD death 105 | Cox | Prospective cohort study |
| Wang 2015 | American black and white people | 24.9, 3.6 | 2177 (45.2%) | 4816 | General population | 27 | Baseline, year-10, 15 and 20. | Uricase method | Fatal and non-fatal CVD. | CVD 164, CHD 79 | | Cox | Prospective cohort study |
| Lai 2016 | Chinese retirees | 62.87, 7.69 | 7190 (44.76%) | 16063 | Elder population without kidney disease | 5 | Baseline and after 5 years. | Colorimetric method | CHD | 1660 | | Cox | Prospective cohort study |
| Zhang 2016 | Japanese | 52.56 | 15628 (43.04%) | 36313 | General population | 10 (average) | Not mentioned. | Phospho-tungstic acid method | CVD death | | CVD death 1288, CHD death 131 | Cox | Prospective cohort study |
| Wu 2017 | Chinese community | 70.7, 5.9 | 1194 (55.74%) | 2142 | Elderly patients without comorbidities | 4.78 (average) | Baseline and every year. | Not mentioned. | CAD events | 213 | CVD death 218 | Poisson regression | Prospective cohort study |
| Andrikou 2018 | Greek | 57.81, 11.67 | 1095 (47.88%) | 2287 | Participants with essential hypertention | 8 (average) | Baseline | Not mentioned. | CAD events | 57 | | Cox | Prospective cohort study |
| Boutet 2020 | Canadian | 53.8 | 8620 (47.5%) | 18149 | General population | 7 | Baseline | Not mentioned. | CVE and CVI | CVE 1944, CVI 381 | | Cox | Prospective cohort study |
| Tian 2020 | Chinese community | 50.3, 12.0 | 55729 (78.00%) | 71449 | Free of MI | 8.96 (average) | Baseline and per 2 years. | Uricase-peroxidase method | MIs | | | Cox | Prospective cohort study |
| Cheng 2021 | Chinese employees | 47.2, 13.9 | 18431 (61.49%) | 29974 | General population | 5.78 (average) | Not mentioned. | Uricase-peroxidase method | CVD | 1062 | | Cox | Prospective cohort study |

*(Continued)*

**Table 1.** (Continued)

| Study | Participant source | Age (mean, SD) | Sex (male) (%) | Sample size | Population feature | Follow-up period (years) | Serum uric acid test time point | Serum uric acid assessment method | Outcome | Number of nonfatal outcomes | Number of fatal outcomes | Statistical model (number of adjusted factors) | Study design |
|---|---|---|---|---|---|---|---|---|---|---|---|---|---|
| Colantonio 2021 | American | Not mentioned. | 353 (42.02%) of the random subcohort | 1926 | General population | 6 | Baseline | Not mentioned. | SCD and CHD | SCD 235, CHD 851 | | Cox | Prospective cohort study |
| Podpalov 2022 | Belarusian | Not mentioned. | Not mentioned. | Not mentioned. | General population | 5 | Baseline | Not mentioned. | MI and CVD death | | Not mentioned. | Multivariate regression model | Prospective cohort study |
| Lee 2023 | Korean | 44.8, 10.5 | 8822 (50.43%) | 17492 | Nondiabetes population with nonCKD or CKD G1-G3a | 4 | Baseline | Not mentioned. | IHD | 335 | | Cox | Prospective cohort study |
| Perticone 2023 | Italian | 52.2, 11.3 | 830 (50.30%) | 1650 | Population with untreated hypertention | 9.5 (average) | Baseline | Uricase-peroxidase method | Coronary events, cerebrovascular events | CHD 250, CVD 118 | | Cox | Prospective cohort study |
| Tian 2023 | Chinese community | 48.12, 12.67 | 16001 (63.29) | 25284 | Population without CVD risk factors and with 3.0–6.0 mg/dl serum UA. | 12.97 (median) | Baseline and per 2 years. | Uricase-peroxidase method | CVD | 1007 | | Cox | Prospective cohort study |
| Wakabayashi 2023 | Japanese | 51.5, 14.2 | 202 (44.89%) | 450 | Obesity outpatient (BMI ≥ 25.0). | 5 | Baseline and after 3 months. | Not mentioned. | CVD and CHD | CVD 39, CHD 15 | | Cox | Prospective cohort study |
| Boyarinova 2024 | Russian | Not mentioned. | Not mentioned. | 4168 | General population | 8 | Baseline | Not mentioned. | CVD | 158 | | Cox | Prospective cohort study |
| Mayo-Juanatey 2025 | Spanish | 61.77, 14.82 | 141 (23.80%) | 591 | Patients with rheumatic diseases. | Not mentioned. | Not mentioned. | Not mentioned. | CVE | Not mentioned. | | Multivariate logistic regression | Prospective cohort study |
| Sarebanhassanabadi 2025 | Iranian | 48.6, 14.7 | 804 (51.80%) | 1552 | General population | 10 | Not mentioned. | Not mentioned. | CAD | 225 | | Cox | Prospective cohort study |

AMI, acute myocardial infarction; AP, angina pectoris; ASCVD, Atherosclerotic Cardiovascular Disease; CAD, coronary artery disease; CHD, coronary heart disease; Cox, Cox proportional hazards multivariate regression model; CVD, cardiovascular disease; CVE, cardiovascular event; CVI, cardiovascular intervention; HF, heart failure; IHD, ischemic heart disease; MI, myocardial infarction; SCD, suddent cardiac death; SD, standard deviation.

evidence: large effect, plausible confounding would change the effect, dose-response relationship. A "Summary of Findings" table were presented to summarize the quality of the evidences for our systematic review.

## Results

### Search results and study characteristics

As Fig 1 shows, we retrieved 2,190 records from five electronic databases. After deduplication, quick screen and full text screen, 42 articles [11–15,17,19–24, 33–62] from 39 studies met the eligible criteria and were included in the systematic review and meta-analysis.

Table 1 and Supplementary Table S1 in S1 Appendix presents the general characteristics of the included studies. The studies were published from 1980 [15] to 2025 [33], with a total sample size of 1,082,880 participants (one conference abstract did not report the sample size [24]). Of these, 613,711 were male (56.89%) and 465,001 were female (43.11%) (one conference abstract of 4,168 participants did not report sex distribution [34]). Six studies included only males [13,17,35,36,37,38], and one study included only females [39]. Sample sizes ranged from 174 [13] to 417,734 [14], with

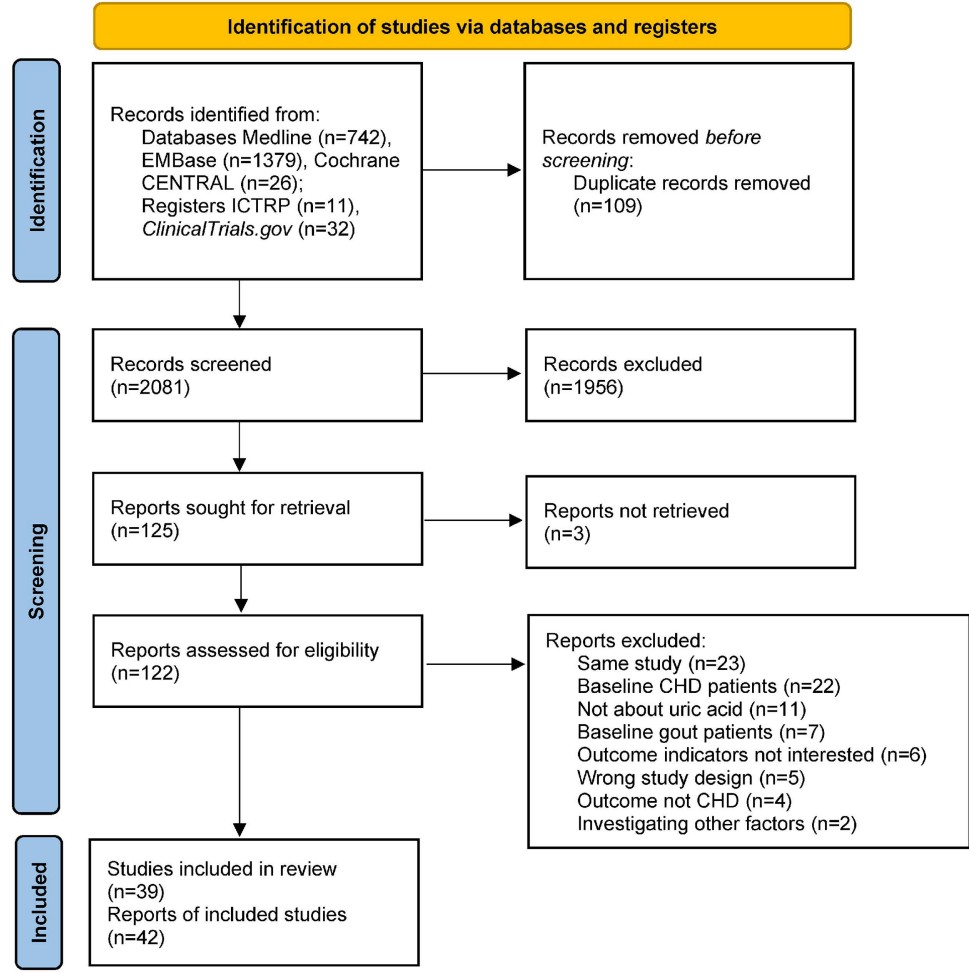

**Fig 1. Flow diagram of the study.**

follow-up durations from 4 [21] to 27 years [40]. Participants were enrolled from North America, Europe, and Asia, with average ages ranged from 37.2 [15] to 70.7 [41] years. Most participants were free of cancer. Some studies included hypertensive participants [42,43,44], Japanese atomic bomb survivors [45], or nondiabetic participants [11,12], etc. Serun UA assessment methods varied: eleven used the uricase-peroxidase method, six used colorimetric method, six used phosphotungstic acid method, five used uricase method, and 14 did not report the methods. Nineteen studies reported explicit hyperuricemia cut-offs, ranged from 5.2–7.0 mg/dL for the general population, 6.25–7.7 mg/dL for males, and 4.6–6.6 mg/dL for females. One study published in 1980 used the "mean+2SD" method to determine hyperuricemia [15]. Seventeen articles reported the results based on quantiles: one tertile [12], eight quartile [14,17,36,46,47,39,48,49,33], seven quintile [35,50,37,51,52,38,53], and one with 8 quantiles [34]. Interestingly, two articles from the same study used quartile [39] and quintile [38] but achieved similar results. Sixteen studies reported the results of per unit increase serum UA, 11 used 1 mg/dL [11,14,23,46,54,55,56,40,41,57,44], and the others varied [22,51,39,58,59]. Thirty-seven articles applied the Cox proportional hazards multivariate regression model, one used the Chi-square analysis [15], one used the Poisson regression [41], and three used multivariate regression [13,24,60]. Among the 39 studies, 37 were prospective cohort study, one retrospective cohort study [11] and one prospective case-control study [13]. The detailed information are presented as Table 1 and Supplementary Table S1 in S1 Appendix.

## Risk of bias of included studies

Fig 2 summarizes the risk of bias. For the "Risk of bias due to confounding" domain, most studies were rated as "Low Risk of Bias, except for concerns about residual confounding", as they adjusted for confounding factors but did not use additional methods (e.g., negative controls) to address residual confounding. Eight studies were rated high risk of bias due to insufficient confounding factors in the regression model [12,14,15,37,55,57,34,49,60]. For the "Risk of bias arising from measurement of the exposure" domain, one study was rated very high risk of bias due to unfounded definition of hyperuricemia cut-off value (5.2 mg/dL) [43], one high risk of bias for not applying the reported cut-off [49], and two some concerns due to slightly arbitrary cut-offs [21,24]. For the "Risk of bias due to post-exposure interventions" domain, one was rated some concerns due to insufficient information from the conference abstract [60]. The other 4 domains of risk of bias were all rated low risk of bias. For the "Overall risk of bias" domain, one study was rated very high risk of bias, eight high risk of bias, and two some concerns according to the highest risk of bias of the 7 domains. One study was rated very high risk of bias due to high risk of bias in domain 1 and domain 2 [49]. Detailed risk of bias assessment is presented as Supplementary Figure S1 in S1 Appendix.

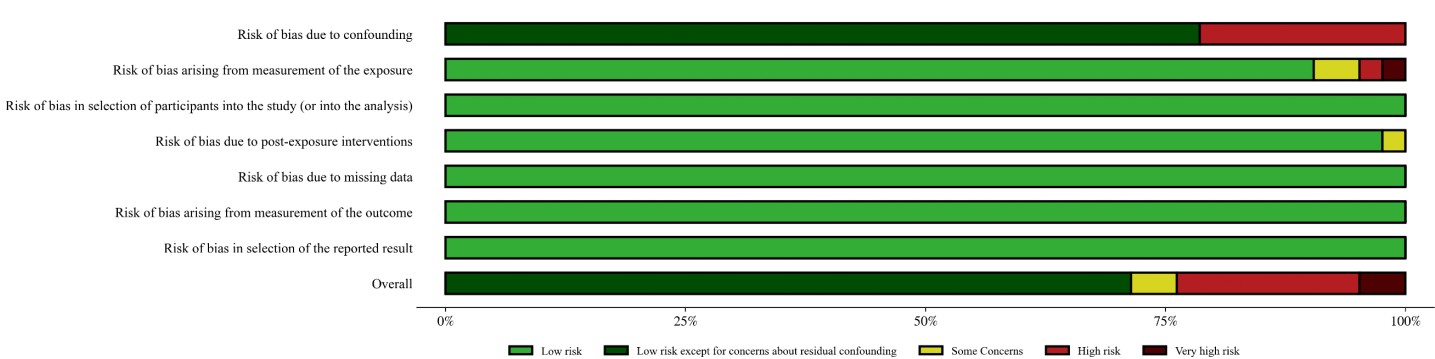

**Fig 2. Risk of bias summary of the included studies.**

## Association between hyperuricemia and cardiovascular outcomes

As Fig 3A presents, a total of 15 studies [12,13,17,20,21,35,50,47,61,51,45,62,41,43,33] were included to investigate the association between hyperuricemia and CHD. The pooled estimation showed significant association between hyperuricemia and CHD [HR 1.21 (95%CI 1.14–1.28), p<0.001, $I^2$=34.34%, $\tau^2$=0], while no significant heterogeneity or publication bias was detected through test. Excluding quantile-based studies [13,20,21,61,45,62,41,43], similar pooled estimation was found [1.24 (95%CI 1.15–1.35), p<0.001, $I^2$=43.09%, $\tau^2$=0] with higher heterogeneity and potential publication bias (Fig 3B). Trim-and-fill test produced a slightly lower but still significant estimation [1.20 (95%CI 1.06–1.35), p=0.003], indicated the reliability that hyperuricemia might increase the risk of CHD.

The secondary outcomes are also summarized as Fig 3A. It is found that hyperuricemia might increase the risk of CHD death [1.20 (1.05–1.36), p=0.005, $I^2$=41.28%, $\tau^2$=0.01], CVD death [1.75 (1.12–2.74), p=0.014, $I^2$=49.48%, $\tau^2$=0.08], and MI [1.23 (1.03–1.47), p=0.025, $I^2$=56.96%, $\tau^2$=0.04] by including 7, 3, and 10 studies in the pooled estimation, respectively. No significant association was found between hyperuricemia and CVD [1.09 (0.94–1.27), p=0.245, $I^2$=0%, $\tau^2$=0], while estimation from trim-and-fill test provided similar result [1.06 (0.92–1.23), p=0.403]. Although no significant publication bias was detected through Egger's test and Begg's test for CHD death, CVD death, and MI, the trim-and-fill test provided controversial results on CHD death [1.16 (0.94–1.41), p=0.16], CVD death [1.37 (0.88–2.14), p=0.168], and MI [1.19 (0.99–1.43), p=0.064].

We further tested the reliability of the secondary outcomes by excluding the quantile data. As Fig 3B shows, only one study was included in the pooled estimations of CHD death and CVD death, respectively. The pooled estimations of 4 studies of CVD [1.58 (1.18–2.11), p=0.002, $I^2$=54.94%, $\tau^2$=0.05] showed increasing risk of hyperuricemia, while the trim-and-fill estimation still showed no significant association [1.31 (0.92–1.87), p=0.135]. The pooled estimation of MI showed stable increasing risk of hyperuricemia [1.10 (1.03–1.18), p=0.004, $I^2$=42.57%, $\tau^2$=0], supported by the trim-and-fill test [1.07 (1.01–1.13), p=0.022].

## Association between serum UA and cardiovascular outcomes, by increasing unit of serum UA

Sixteen studies reported the association between cardiovascular outcomes and the serum UA by increasing unit, 11 [11,14,23,46,54,55,56,40,41,57,44] with 1 mg/dL and five with other units [22,51,39,58,59]. Increasing risks of unit increase of serum UA were found for all cardiovascular outcomes (Fig 4A), including CHD [1.16 (1.05–1.29), p=0.005, $I^2$=91.25%, $\tau^2$=0.02], CHD death [1.13 (1.09–1.17), p<0.001, $I^2$=43.22%, $\tau^2$=0], CVD [1.12 (1.04–1.21), p=0.004, $I^2$=84.78%, $\tau^2$=0.01], CVD death [1.11 (1.09–1.13), p<0.01, $I^2$=16.94%, $\tau^2$=0], and MI [1.07 (1.06–1.09), p<0.001, $I^2$=0%, $\tau^2$=0]. Potential publication bias was detected through Egger's test and Begg's test for CVD, and the trim-and-fill test showed no significant association between unit increase of serum UA and CVD [1.03 (0.92–1.16), p=0.584]. Similar results were found after including only the 1 mg/dL data (Fig 4B), that the increasing serum UA of 1 mg/dL increased the risk of CHD [1.16 (1.02–1.31), p=0.022, $I^2$=92.9%, $\tau^2$=0.02], CHD death [1.14 (1.08–1.20), p<0.001, $I^2$=52.09%, $\tau^2$=0], CVD [1.13 (1.02–1.25), p=0.017, $I^2$=84.30%, $\tau^2$=0.01], CVD death [1.12 (1.09–1.16), p<0.001, $I^2$=34.6%, $\tau^2$=0], and MI [1.07 (1.06–1.09), p<0.001], and potential publication bias was detected for CVD [trim-and-fill 1.03 (0.90–1.17), p=0.704].

## Dose-response meta-analysis

As above-mentioned, 17 studies with quantile data were included in the dose-response meta-analysis. Among the 5 cardiovascular outcomes, serum UA showed linear association with CHD and CVD, while nonlinear association was found with CHD death, CVD death, and MI. Significant dose-response effect was found for an increase of 1 mg/dL serum UA on CVD risk [linear dose-response 1.08 (1.03–1.14), p=0.003, $I^2$=0%]. However, no significant dose-response effect was found on CHD, CHD death, CVD death, and MI (Fig 5A).

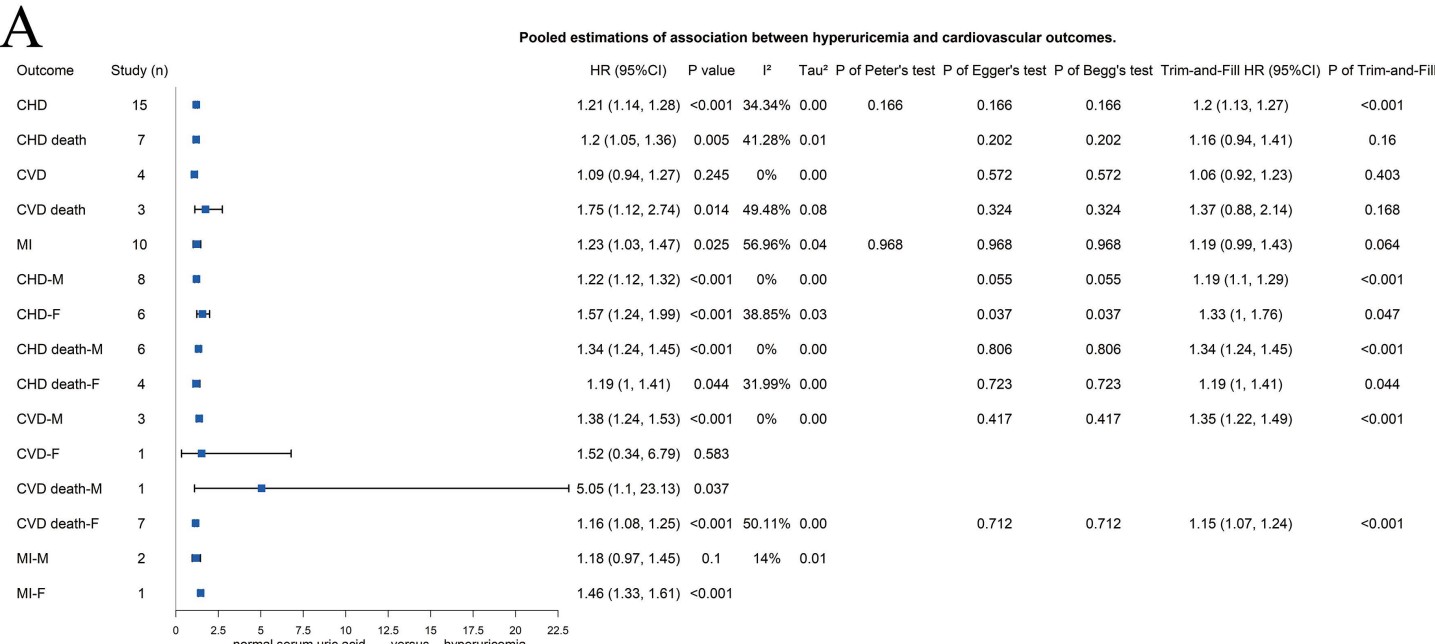

A **Pooled estimations of association between hyperuricemia and cardiovascular outcomes.**

| Outcome | Study (n) | HR (95%CI) | P value | I² | Tau² | P of Peter's test | P of Egger's test | P of Begg's test | Trim-and-Fill HR (95%CI) | P of Trim-and-Fill |
|---|---|---|---|---|---|---|---|---|---|---|
| CHD | 15 | 1.21 (1.14, 1.28) | <0.001 | 34.34% | 0.00 | 0.166 | 0.166 | 0.166 | 1.2 (1.13, 1.27) | <0.001 |
| CHD death | 7 | 1.2 (1.05, 1.36) | 0.005 | 41.28% | 0.01 | | 0.202 | 0.202 | 1.16 (0.94, 1.41) | 0.16 |
| CVD | 4 | 1.09 (0.94, 1.27) | 0.245 | 0% | 0.00 | | 0.572 | 0.572 | 1.06 (0.92, 1.23) | 0.403 |
| CVD death | 3 | 1.75 (1.12, 2.74) | 0.014 | 49.48% | 0.08 | | 0.324 | 0.324 | 1.37 (0.88, 2.14) | 0.168 |
| MI | 10 | 1.23 (1.03, 1.47) | 0.025 | 56.96% | 0.04 | 0.968 | 0.968 | 0.968 | 1.19 (0.99, 1.43) | 0.064 |
| CHD-M | 8 | 1.22 (1.12, 1.32) | <0.001 | 0% | 0.00 | | 0.055 | 0.055 | 1.19 (1.1, 1.29) | <0.001 |
| CHD-F | 6 | 1.57 (1.24, 1.99) | <0.001 | 38.85% | 0.03 | | 0.037 | 0.037 | 1.33 (1, 1.76) | 0.047 |
| CHD death-M | 6 | 1.34 (1.24, 1.45) | <0.001 | 0% | 0.00 | | 0.806 | 0.806 | 1.34 (1.24, 1.45) | <0.001 |
| CHD death-F | 4 | 1.19 (1, 1.41) | 0.044 | 31.99% | 0.00 | | 0.723 | 0.723 | 1.19 (1, 1.41) | 0.044 |
| CVD-M | 3 | 1.38 (1.24, 1.53) | <0.001 | 0% | 0.00 | | 0.417 | 0.417 | 1.35 (1.22, 1.49) | <0.001 |
| CVD-F | 1 | 1.52 (0.34, 6.79) | 0.583 | | | | | | | |
| CVD death-M | 1 | 5.05 (1.1, 23.13) | 0.037 | | | | | | | |
| CVD death-F | 7 | 1.16 (1.08, 1.25) | <0.001 | 50.11% | 0.00 | | 0.712 | 0.712 | 1.15 (1.07, 1.24) | <0.001 |
| MI-M | 2 | 1.18 (0.97, 1.45) | 0.1 | 14% | 0.01 | | | | | |
| MI-F | 1 | 1.46 (1.33, 1.61) | <0.001 | | | | | | | |

normal serum uric acid    versus    hyperuricemia

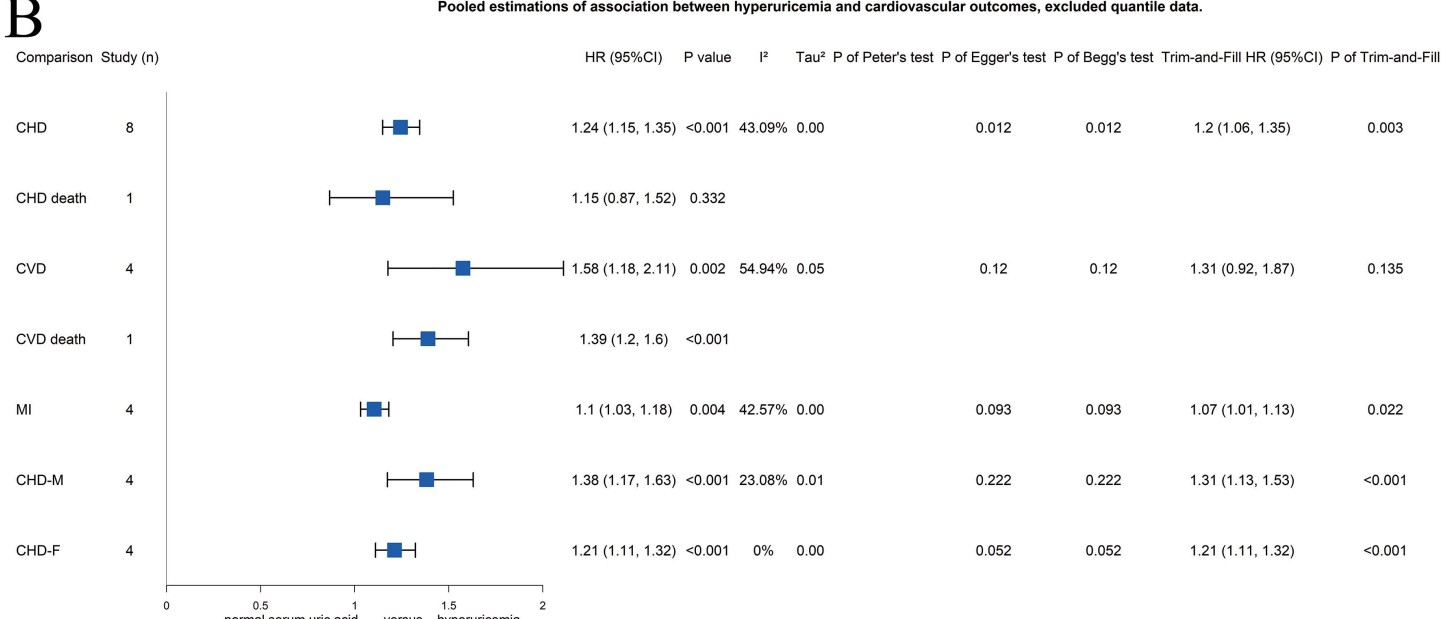

B **Pooled estimations of association between hyperuricemia and cardiovascular outcomes, excluded quantile data.**

| Comparison | Study (n) | HR (95%CI) | P value | I² | Tau² | P of Peter's test | P of Egger's test | P of Begg's test | Trim-and-Fill HR (95%CI) | P of Trim-and-Fill |
|---|---|---|---|---|---|---|---|---|---|---|
| CHD | 8 | 1.24 (1.15, 1.35) | <0.001 | 43.09% | 0.00 | | 0.012 | 0.012 | 1.2 (1.06, 1.35) | 0.003 |
| CHD death | 1 | 1.15 (0.87, 1.52) | 0.332 | | | | | | | |
| CVD | 4 | 1.58 (1.18, 2.11) | 0.002 | 54.94% | 0.05 | | 0.12 | 0.12 | 1.31 (0.92, 1.87) | 0.135 |
| CVD death | 1 | 1.39 (1.2, 1.6) | <0.001 | | | | | | | |
| MI | 4 | 1.1 (1.03, 1.18) | 0.004 | 42.57% | 0.00 | | 0.093 | 0.093 | 1.07 (1.01, 1.13) | 0.022 |
| CHD-M | 4 | 1.38 (1.17, 1.63) | <0.001 | 23.08% | 0.01 | | 0.222 | 0.222 | 1.31 (1.13, 1.53) | <0.001 |
| CHD-F | 4 | 1.21 (1.11, 1.32) | <0.001 | 0% | 0.00 | | 0.052 | 0.052 | 1.21 (1.11, 1.32) | <0.001 |

normal serum uric acid    versus    hyperuricemia

**Fig 3. Forest plots of association between hyperuricemia and risk of coronary heart disease and other cardiovascular outcomes of (A) all included studies or (B) without studies of quantile data.** CHD, coronary heart disease; CVD, cardiovascular disease; F, female; HR, hazard ratio; MI, myocardial infraction; M, male.

## Subgroup analysis

We performed subgroup analysis for all pooled estimations based on sex. As Fig 3A and Supplementary Figure S2 in S1 Appendix present, hyperuricemia significantly increased the risk of CHD, CHD death, and CVD death among both male and female population, while increased the risk of CVD among male population and MI among female population. Fig 3B

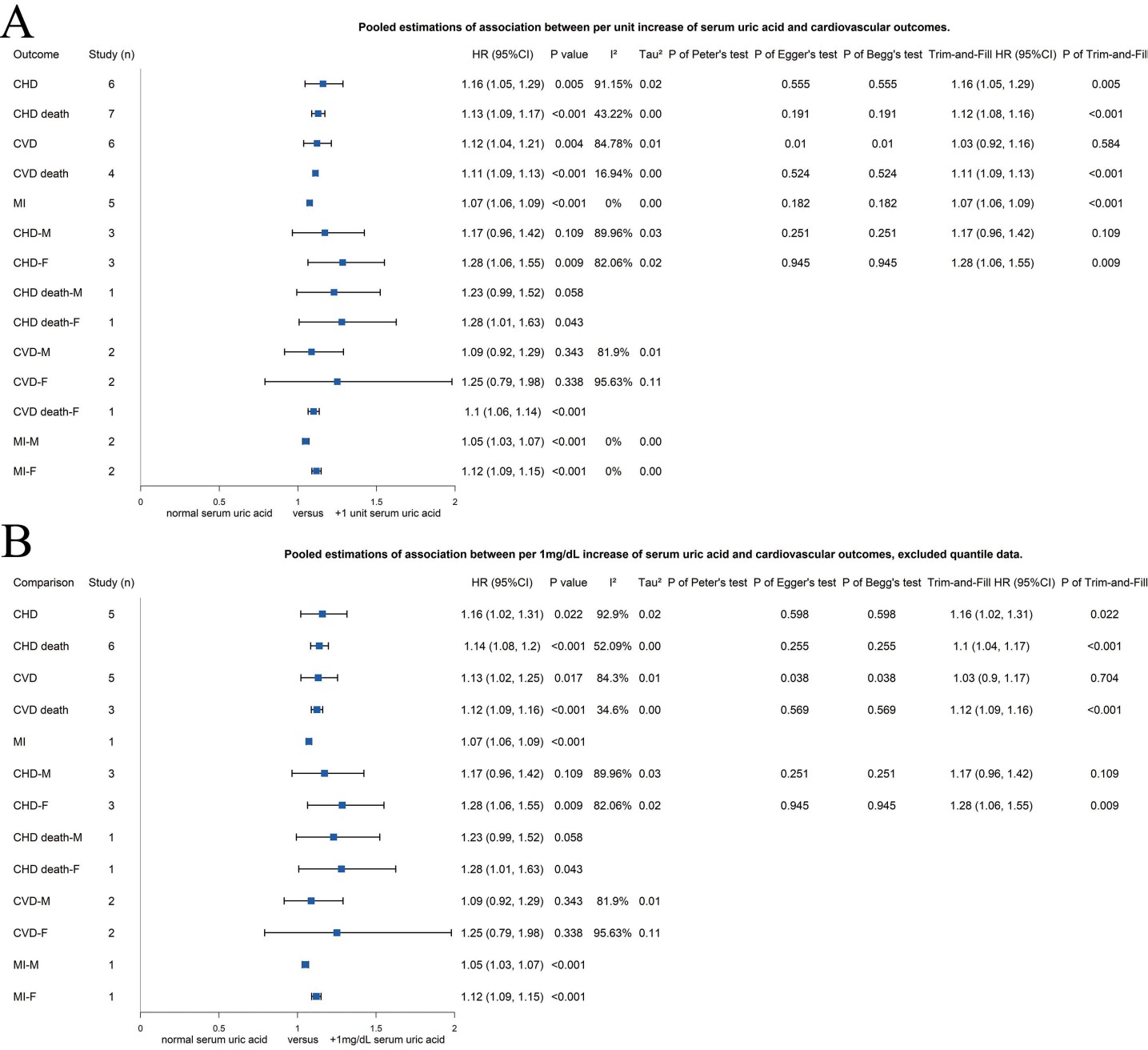

**Fig 4. Forest plots of association between risk of coronary heart disease and other cardiovascular outcomes and (A) all included studies with data of increased unit serum uric acid or (B) studies with data of increased 1 mg/dL serum uric acid.** CHD, coronary heart disease; CVD, cardiovascular disease; F, female; HR, hazard ratio; MI, myocardial infraction; M, male.

and Supplementary Figure S2 in S1 Appendix present the pooled estimations without quantile data, and found significantly increasing risk of CHD among both male and female population from hyperuricemia.

As Fig 4A and Supplementary Figure S3 in S1 Appendix present, the risk of CHD, CHD death, and CVD death from female population, and MI from both male and female population increased for each increase of 1 unit serum UA; as

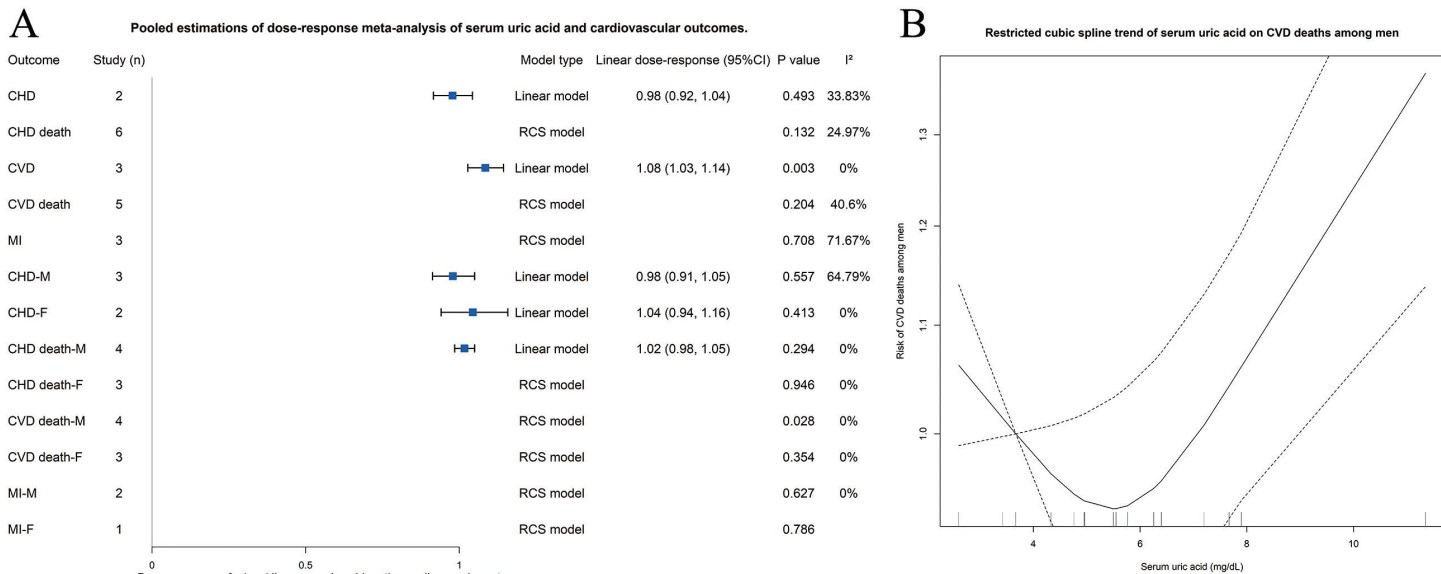

**Fig 5. (A)** Forest plot of dose-response meta-analysis of association between serum uric acid and risk of coronary heart disease and other cardiovascular outcomes. **(B)** Line graph with confidence interval of restricted cubic spline of association between serum uric acid and risk of cardiovascular disease death among male population. CHD, coronary heart disease; CVD, cardiovascular disease; F, female; HR, hazard ratio; MI, myocardial infraction; M, male.

considering for each increase of 1 mg/dL serum UA, only studies on MI were excluded, and the risk of MI from both male and female population still increased for each increase of 1 mg/dL serum UA.

To explore the impact of different statistics (HR or RR) used in the original studies on the results of the meta-analysis, as well as the potential sources of heterogeneity, we conducted subgroup analyses by statistics. As shown in Supplementary Table S2 in S1 Appendix, for the association between hyperuricemia and the risk of CHD and other cardiovascular outcomes, the pooled estimates in the HR subgroup were largely consistent with the overall results, with markedly lower heterogeneity observed in the CHD death and CVD death comparisons. In contrast, the pooled results in the RR subgroup differed significantly from the overall estimations and exhibited substantial heterogeneity. Regarding the association between each unit increase in serum UA and the risk of CHD and other cardiovascular outcomes, as shown in Supplementary Table S3 in S1 Appendix, both the HR and RR subgroup analyses yielded results generally consistent with the overall findings, while the CHD death and CVD death subgroups again demonstrated notably lower heterogeneity. Due to the limited number of included studies, heterogeneity could not be assessed in the RR subgroup meta-analysis.

We also performed subgroup analysis based on sex for dose-response meta-analysis. As Fig 5A shows, 3 subgroups were analyzed with linear model and the others with RCS model, while significant dose-response effect was found on CVD death among male population, indicated a U-shaped trend that increase of serum UA would lead to the decreased risk of CVD death when serum UA was under approximately 4.5 mg/dL and lead to the increase risk of CVD death afterwards among male population (Fig 5B).

## Meta-regression

We performed meta-regression to identify potential factors that might be associated with the risk of cardiovascular outcomes. However, due to the incomplete data reported by the included studies, limited factors were involved in the meta-regression. As shown in Supplementary Table S4 in S1 Appendix, when age, sex, and BMI were included as

covariates in the meta-regression, only the data for CHD death and MI were analyzable. Male sex was significantly associated with a lower risk of CHD death (β = −0.91, 95%CI: −1.66 to −0.15, p = 0.018). When only age and sex were included as covariates, the association between male sex and CHD death was not significant, whereas male sex was significantly associated with a slightly lower risk of CVD (β = 0, 95%CI: −0.01 to 0, p = 0.043). As shown in Supplementary Table S5 in S1 Appendix, neither age nor sex was associated with the risk of CHD or other cardiovascular outcomes by per unit increase in serum UA.

### Sensitivity analysis

We performed sensitivity analysis for all pooled pair-wise estimations when included more than 1 study. As Supplementary Figure S2 and S3 in S1 Appendix presents, the pooled estimations were stable with leave-one-out sensitivity analysis. However, possibly due to the relatively small number of included studies, some pooled estimations of subgroup analysis were shaken in sensitivity analysis.

### Quality of the evidence

We evaluated the quality of evidence of each cardiovascular outcome according to the guidance of the GRADE recommendation (Table 2). Among the 5 cardiovascular outcomes, CHD as the primary outcome indicated low quality of evidence, without downgraded or upgraded. All of the other 4 outcomes indicated very low quality of evidence. The CVD was downgraded for potential serious risk of bias (2 high risk of bias among included 6 studies). Another 3 outcomes, including CHD death, CVD, and MI, were downgraded for potential inconsistency, which was possibly due to quantile data. All of the outcomes were not downgraded for indirectness or imprecision, while CVD death and MI were downgraded for potential publication bias, detected by trim-and-fill test. In addition, CVD was upgraded for significant linear dose-response effect.

**Table 2. GRADE evidence profile of association between hyperuricemia and cardiovascular outcomes.**

| Outcome | Downgrade | | | | | Upgrade | | | Quality of the evidence (GRADE) |
|---------|-----------|---|---|---|---|---------|---|---|---------------------------------|
| | **Risk of bias** | **Inconsistency** | **Indirectness** | **Imprecision** | **Publication bias** | **Large effect** | **Dose-response** | **All plausible confounding would reduce the effect** | |
| CHD | No serious risk of bias | No serious inconsistency | No serious indirectness | No serious imprecision | No serious publication bias | No large effect | Not significant | Not reduce | ⊕⊕⊖⊖ Low |
| CHD death | No serious risk of bias | Potential inconsistency, possibly due to quantile data [1] | No serious indirectness | No serious imprecision | Potential publication bias[2] | Not reduce | Not significant | Not reduce | ⊕⊖⊖⊖ Very low |
| CVD | Potential serious risk of bias[3] | Potential inconsistency, possibly due to quantile data [1] | No serious indirectness | No serious imprecision | No serious publication bias | Not reduce | Significant linear dose-response effect[4] | Not reduce | ⊕⊖⊖⊖ Very low |
| CVD death | No serious risk of bias | No serious inconsistency | No serious indirectness | No serious imprecision | Potential publication bias[1] | Not reduce | Not significant | Not reduce | ⊕⊖⊖⊖ Very low |
| MI | No serious risk of bias | Potential inconsistency, possibly due to quantile data[1] | No serious indirectness | No serious imprecision | Potential publication bias[2] | Not reduce | Not significant | Not reduce | ⊕⊖⊖⊖ Very low |

1 Downgrade for significant difference between the pooled estimation by including or excluding the quantile data.

2 Downgrade for significant difference between pooled estimation and trim-and-fill estimation (p < 0.05 versus p > 0.05).

3 Downgrade for 2 of 6 included studies with overall high risk of bias.

4 Upgrade for dose-response meta-analysis linear model p < 0.05, suggests potential dose-response effect.

5 Not downgrade for $I^2 > 50\%$ in the pooled estimation, while $I^2 < 50$ after excluding the quantile data.

## Discussion

Our systematic review and meta-analysis, including 42 articles from 39 studies with a total of 1,082,880 participants, found that hyperuricemia is associated with an increased risk of CHD and other cardiovascular outcomes, including CHD death, CVD death, and MI. This association was supported by analyses evaluating risk of per unit increase in serum UA. The findings remained consistent in most subgroups. Dose-response meta-analysis revealed a linear association between serum UA and CVD risk and a U-shaped association between serum UA and CVD death in men. The overall quality of evidence was low for CHD and very low for the other cardiovascular outcomes.

The potential association between UA levels and the development of CHD and other CVDs from a growing body of experimental and clinical evidence suggesting that hyperuricemia may play a critical role in the pathophysiology of atherosclerosis and endothelial dysfunction. Emerging studies have demonstrated that elevated UA not only serves as a marker of metabolic disturbance but may also exert direct detrimental effects on vascular health [63]. For instance, UA has been shown to promote oxidative stress and inflammation, processes that are pivotal in the development of atherosclerosis [64]. Experimental models indicate that high levels of UA can stimulate the production of reactive oxygen species (ROS), which leads to endothelial injury and enhances the expression of adhesion molecules [65]. This subsequently promotes monocyte adhesion and infiltration into the vascular wall, fostering the development of atherosclerotic plaques [66]. Moreover, laboratory studies have revealed that UA can induce the proliferation of vascular smooth muscle cells (VSMCs) and foam cell formation [67], all of which contribute to plaque instability and, ultimately, acute coronary events. Additionally, animal studies have elucidated mechanisms by which hyperuricemia disrupts nitric oxide (NO) signaling, inhibiting vasodilation and promoting vascular stiffness [63]. The accumulation of UA may also correlate with metabolic syndrome components, such as obesity and insulin resistance [68], further compounding cardiovascular risk. Overall, these experimental findings support the hypothesis that elevated serum UA levels can contribute to the onset and progression of CHD and other CVDs through multiple interrelated pathways, including oxidative stress, inflammation, and vascular remodeling [69]. Thus, unraveling the mechanistic links between UA and cardiovascular health is crucial for identifying potential therapeutic targets and improving patient outcomes in the management of CVDs.

In a recent comparative systematic review [70], a more pronounced increase in risk associated with hyperuricemia was reported compared to the findings of the present study. This review highlighted a significant association between high serum UA levels and the elevated risk of CHD and CVD in the general population, with larger effect size compared with our study. Although the reference information of the included studies was not reported in this systematic review [70], we tried to identify their included studies and found some important studies omitted (e.g., Holme et al., 2009 [14]); furthermore, the inclusion of some studies might lead to serious bias, particularly those involving pharmacological interventions [71]. Such omissions and inclusions may substantially increase the risk of bias of their conclusion, which could account for the discrepancies between their findings and those of our study. Another systematic review published in 2018 [72], which analyzed data from 1,134,073 subjects and assessed the association between serum UA and CVD death risk, found a significant positive correlation between UA levels and cardiovascular death risk. They reported similar results with our study. However, this systematic review [72] did not limit inclusion criteria based on whether participants were free of CVD or CHD at baseline, potentially resulting in greater heterogeneity in the included studies ($I^2$: 79% versus 49.48% in our study). Additionally, a systematic review published in 2016 identified an association between hyperuricemia and increased incidence and mortality from CHD, particularly in female populations [10]. Although this finding is consistent with another systematic review published in the same year [16], it remains a subject of contention. Some researchers have raised concerns regarding the robustness of this conclusion, suggesting that it may be influenced by insufficient adjustment for confounding variables in certain studies included in the review [9]. Consequently, they argue that the association between hyperuricemia and increased risk of incidence and mortality from CHD is not definitively established.

Our study identified a U-shaped association between serum UA levels and the risk of CVD mortality among men. As shown in Supplementary Figure S4 in S1 Appendix, a similar U-shaped pattern was also observed for CHD mortality in

men (p > 0.05), whereas the risk of CHD itself demonstrated an inverted U-shaped association (p > 0.05). Interestingly, these associations were not observed among women. Similar pattern has been observed from other studies as well, including kidney failure and mortality in chronic kidney disease [73], composite outcome of COVID-19 patients [74], etc. The relationship between hyperuricemia and endothelial dysfunction or vascular injury has been well recognized [75]. Some studies have suggested that UA possesses physiological antioxidant [76] and immunomodulatory properties [77], and that excessively low UA levels may impair free radical scavenging capacity and reduce immune regulation, ultimately leading to endothelial damage. This mechanism might explain the increased CVD mortality observed in men with low UA levels. However, it does not account for the inverted U-shaped association with CHD risk in men, nor the absence of similar patterns in women. It should be acknowledged that the number of original studies included in the dose-response meta-analysis was limited, which may affect the stability of the results. Further studies with high-quality are warranted to verify or correct our findings. We also anticipate that future research on UA will provide a more comprehensive understanding of its dual effects at different concentration levels.

Previous systematic review indicated substantial heterogeneity among published studies [8], which may compromise the reliability of their conclusions. The sources of heterogeneity include irreducible demographic characteristics of the included populations, as well as variations in the cut-off values used to define hyperuricemia (including the analysis of quantile data) and the confounding factors involved in the statistical models. Consistent with previous findings [9], our study also identifies an increasing recognition of risk factors influencing the incidence and mortality of CHD and CVD over time [78]. Although the confounding factors included in statistical models have not remained entirely consistent across recent studies, recent studies have involved more and more comprehensive confounding factors (see Table S1 in S1 Appendix), thereby enhancing the reliability of the association between hyperuricemia and the risks of CHD and other CVDs. Our findings reveal that the differences in results between studies accounting for varying degrees of confounding factors are not substantial, and the heterogeneity among studies is generally low (see Fig 3 and Fig 4), allowing for robust conclusions in meta-analyses. This stability may be attributed to the increasing sample sizes, extending follow-up periods, and the more consistent baseline characteristics of participants, which collectively reduce the potential influence of random bias on study outcomes, thereby yielding more reliable results of the original studies. Furthermore, variations in the quantile thresholds and the diagnostic cut-off values for hyperuricemia across different studies could significantly impact future research on this condition. Several studies have proposed that the cut-off for serum UA concerning CHD and CVD should not coincide with the hyperuricemia threshold [79–81]. It should, rather, be set below the current cut-off value for the diagnose of hyperuricemia, though consensus on this point remains elusive. This disparity suggests that serum UA may act as a risk factor for CHD and CVD even prior to reaching hyperuricemia or gout. Notably, some studies indicate that the diagnostic threshold for hyperuricemia in women is lower by 1 mg/dL compared to men, and results have frequently shown a stronger association between hyperuricemia and the increased risks of CHD and CVD in female populations rather than in male population, potentially supporting this hypothesis, which seems to be consistent with our meta-regression. In addition to sex, we also conducted subgroup analyses by separating studies that reported results using HRs and those using RRs. We found that heterogeneity was substantially reduced in several outcomes reported with HRs, whereas no such reduction was observed in RR-reported results. This partially reveals one of the potential sources of heterogeneity. Notably, as shown in Supplementary Table S1 in S1 Appendix, all studies published in the past decade have reported their results using HRs, which may help to partially control inter-study heterogeneity in the further systematic reviews. Furthermore, we observed considerable variation in how different studies defined the same cardiovascular outcomes. For example, CVD could refer to any combination of fatal or nonfatal CHD, MI, stroke, or other vascular conditions in different original studies. There were also discrepancies in disease coding systems, ranging from ICD-7 to ICD-10. Additionally, the methods used to determine whether an outcome event occurred varied across studies, including assessments based on medical records, self-reports by participants or family members, or clinical symptoms and auxiliary examinations. Decisions were made by different numbers of investigators or specialized committees in different original

studies. Besides, the measurement of serum UA also differed among studies, employing diverse methods such as the uricase–peroxidase method, colorimetric method, phosphotungstic acid method, uricase method, or unreported techniques. The differences in measurement methods and cut-off values for defining hyperuricemia inevitably contributed to between-study heterogeneity. Finally, some sources of heterogeneity may be difficult to avoid, such as variations in study region, human race, sex distribution, age, and BMI, all of which can introduce residual and unresolvable heterogeneity. We recommend that future original studies take these factors fully into consideration and strive to establish more standardized research protocols and reporting criteria, such as unified cutoff values for hyperuricemia, standardized methods for UA measurement, consistent definitions and adjudication criteria for cardiovascular outcomes, and the use of standardized statistical models and covariates. Such efforts would reduce heterogeneity and facilitate a more reliable and persuasive elucidation of the relationship between hyperuricemia and CHD or other cardiovascular outcomes.

The original studies included in this research excluded participants with CHD or other CVDs at baseline, thereby strengthened the reliability of the association between serum UA and the risk of CHD or CVD. A systematic review [82] has addressed this issue, revealing that MI survivors with elevated serum UA exhibited an increased incidence of major adverse cardiovascular events (MACE) during their hospitalization. Furthermore, these individuals demonstrate higher rates of in-hospital mortality and mortality within one-year post-hospitalization [82]. Another systematic review found a positive correlation between serum UA and the risk of adverse events in patients with chronic heart failure [83]. In hypertensive populations, elevated UA was significantly associated with increased risks of cardiovascular mortality, all-cause mortality, CHD, and MACE [84]. While establishing a causal association between UA and CHD or other cardiovascular outcomes remains challenging, UA or gout have been identified as important potential biomarkers for CHD prognosis [85], suggesting that urate-lowering treatment may confer benefits. A systematic review involved 85,926 participants found that urate-lowering therapy significantly reduced all-cause mortality in patients with gout or hyperuricemia, although specific drug had no significant effect on cardiovascular-specific mortality [86]. To explore the safety and differences among treatments, studies have summarized the cardiovascular safety of urate-lowering therapies for patients with gout or hyperuricemia, revealing that febuxostat was associated with a higher risk of arrhythmia compared to allopurinol; overall, urate-lowering therapies appeared to demonstrate relatively good cardiovascular safety in these patients [87]. Another meta-analysis involving 3,803,509 participants indicated that xanthine oxidase inhibitors were not associated with a reduction in cardiovascular events compared to placebo, and febuxostat may reduce the risk of heart failure compared to xanthine oxidase inhibitors [88]. Collectively, these studies suggested that hyperuricemia poses significant risks to both affected and unaffected populations. While lowering serum UA generally appears safe, there are notable differences in efficacy and safety among various medications, warranting further investigation. Additionally, some studies have indicated that low serum UA may be associated with increased all-cause mortality, cause-specific mortality, and cardiovascular risk [89,90], underscoring the need for future therapeutic research to focus on determining the optimal therapeutic range for serum UA. Currently, there is a substantial body of research on urate-lowering therapy for gout [91], but studies specifically addressing the management of asymptomatic hyperuricemia remain limited. Our study reports that hyperuricemia is associated with an increased risk of cardiovascular outcomes, yet it remains unclear whether urate-lowering interventions in this high-risk population would confer cardiovascular benefits, and which urate-lowering therapies would be better. We encourage researchers to conduct large-scale, long-term cohort studies in this field to evaluate the urate-lowering effects of various pharmacological and non-pharmacological interventions and to assess their long-term impact on cardiovascular outcomes in high-risk populations, thereby providing evidence for future clinical practice.

Using the GRADE tool, the quality of evidence was rated as low for CHD and very low for the other cardiovascular outcomes. It is important to emphasize that, due to the inherent limitations of observational studies, such as susceptibility to confounding factors and biases, they cannot control for bias through randomization as randomized controlled trials (RCTs) do. Therefore, the initial level of evidence for observational studies starts at low rather than high [92]. In our systematic

review, several original studies were rated as having a high risk of bias, and substantial unexplained heterogeneity was also observed as above-mentioned. As described earlier, subgroup analyses and meta-regressions were conducted but could not adequately account for the heterogeneity. Although we performed a dose–response meta-analysis and other complementary analyses, residual confounding between original studies could not be fully eliminated, and the quality of evidence could not be upgraded. To our knowledge, this systematic review is the first to apply the GRADE framework to evaluate the quality of evidence regarding the association between hyperuricemia and CHD and other cardiovascular outcomes. Our findings highlight the need for cautious interpretation of the observed association between hyperuricemia and the clinical outcomes. Future original studies with higher methodological quality are warranted to provide more robust evidence to support clinical practice.

This study presents several notable strengths. Our systematic review specifically limited the inclusion of subjects without a history of CHD, CVD, or gout at baseline, thereby reducing inter-study heterogeneity and enhancing the reliability of the results. We comprehensively summarized all eligible studies, incorporating multiple large-sample, long-term prospective cohort studies, and provided a thorough overview of the involved confounding factors utilized in the statistical models of each original study, which contributes to the robustness of our findings. According to the methods of previous systematic reviews in this field, we incorporated quantile data into the meta-analysis to improve statistical power, subsequently performing a subgroup analysis by excluding quantile data to validate the reliability of our statistical results. Additionally, we calculated the risks of CHD or CVD based on each unit increase in serum UA and conducted a dose-response meta-analysis utilizing quantile data, further supporting our research outcomes. We employed various methods to assess heterogeneity, detect publication bias, and evaluate the impact of such bias on our results. Furthermore, we conducted subgroup analyses based on sex, and performed sensitivity analyses for each pooled estimation, thereby providing a more comprehensive understanding of the findings. Ultimately, we also assessed the grade of evidence for our conclusions.

However, this systematic review has limitations. First of all, despite our efforts to obtain all relevant literature, certain studies remain inaccessible [93,94]. Secondly, substantial and potentially irreducible heterogeneity may exist among the included studies due to variations in the definitions of identical outcome measures, differences in diagnostic criteria for the same disease, updates in diagnostic codes, and discrepancies in how outcome events were ascertained (see Supplementary Table S1 in S1 Appendix). These factors are consistent with the heterogeneity observed in our data. For the same reason, our results may underestimate the prevalence of CHD and CVD, given that the majority of these conditions were identified through self-reports, medical records, and/or death certificates, as well as electrocardiograms and serum cardiac enzymes in original studies; consequently, there may be a considerable number of asymptomatic cases of CHD and CVD that were overlooked, which highlights the necessity for advancements in non-invasive diagnostic methods in the future. Additionally, while we conducted a thorough search of major databases and included non-English articles, one study was translated via Google Translate [13], we may still have inadvertently overlooked other non-English studies that are not indexed in the databases we accessed.

## Conclusion

This systematic review and dose-response meta-analysis provides evidences of low or very low quality, that hyperuricemia might be associated with increased risk of CHD, and other fatal or nonfatal CVDs. Further studies should explore the optimal serum UA threshold to identify the risk of CHD and CVD, and comprehensively include confounding factors to avoid potential bias.

## Supporting information

**S1 Appendix. Supplementary Materials.**
(PDF)

**S2 Checklist. PRISMA Checklist.**
(PDF)

**S3 Checklist. MOOSE Checklist.**
(PDF)

## Author contributions

**Conceptualization:** Diyang Lyu, Liyong Ma, Jingen Li, Yong Zhang.

**Data curation:** Diyang Lyu, Rui Zhuang, Jiaqi Li, Jingen Li, Yong Zhang.

**Formal analysis:** Diyang Lyu.

**Funding acquisition:** Diyang Lyu, Jingen Li.

**Investigation:** Diyang Lyu, Yong Zhang.

**Methodology:** Diyang Lyu, Rui Zhuang, Jiaqi Li, Jingen Li, Yong Zhang.

**Project administration:** Diyang Lyu.

**Resources:** Diyang Lyu.

**Software:** Diyang Lyu, Yucen Wu, Yiming Di, Meifen Song.

**Supervision:** Jingen Li, Yong Zhang.

**Validation:** Diyang Lyu.

**Visualization:** Yucen Wu, Yiming Di, Meifen Song.

**Writing – original draft:** Diyang Lyu.

**Writing – review & editing:** Jiaqi Li, Jingen Li, Yong Zhang.

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
