## [Decision Letter · Decision Letter 0]

30 Sep 2025

PONE-D-25-42229Association of hyperuricemia with coronary heart disease and other cardiovascular outcomes: A systematic review and dose-response meta-analysisPLOS ONE

Dear Dr. Lyu,

Thank you for submitting your manuscript to PLOS ONE. After careful consideration, we feel that it has merit but does not fully meet PLOS ONE’s publication criteria as it currently stands. Therefore, we invite you to submit a revised version of the manuscript that addresses the points raised during the review process.

We look forward to receiving your revised manuscript.

Kind regards,

Andres Mauricio Acevedo-Melo, M.D.

Academic Editor

PLOS ONE

Journal Requirements:

2. Please remove all personal information, ensure that the data shared are in accordance with participant consent, and re-upload a fully anonymized data set.

Additional guidance on preparing raw data for publication can be found in our Data Policy (https://journals.plos.org/plosone/s/data-availability#loc-human-research-participant-data-and-other-sensitive-data ) and in the following article: http://www.bmj.com/content/340/bmj.c181.long .

Additional Editor Comments :

In this manuscript, Lyu et al. present a systematic review and meta-analysis of observational studies evaluating the association between hyperuricemia and coronary heart disease (CHD). By systematically searching Medline, the Cochrane Library, Embase, and two trial registration databases from inception to June 30, 2025, the authors identified 42 articles corresponding to 39 unique studies, encompassing a total of 1,082,880 participants. The pooled results demonstrated a statistically significant association between hyperuricemia and increased risk of CHD, as well as its individual components (CHD death, cardiovascular death, myocardial infarction). The dose–response analyses further suggested a linear relationship between serum uric acid (UA) and cardiovascular disease (CVD) risk, and a U-shaped association between serum UA and risk of CVD death among men. Despite the relevance of the research question and the adherence to a registered protocol, the overall certainty of the evidence was limited by the observational study designs. Several key aspects require clarification and further development before the manuscript can be considered suitable for publication.

Risk of Bias: Although the original protocol did not specify separate analyses for CHD-related outcomes, the authors expanded their work to include CHD death, CVD death, and myocardial infarction. While this approach is wisely comprehensive, it raises concerns regarding potential spin bias in relation to the primary outcome (CHD). Importantly, substantial heterogeneity was observed across included studies, arising from differences in association measures, uric acid assessment methods, and other study-level factors. The sources of this statistical heterogeneity should be explicitly examined and reported in relation to study characteristics and association measurements (HR,OR,RR). Furthermore, given the relevance of confounding factors in the relationship between hyperuricemia and CHD, meta-regression analyses should be performed at least for age, sex, and body mass index (BMI).

Discussion: The Discussion would benefit from additional depth. Specifically, the clinical relevance of the findings should be contextualized in light of potential therapeutic strategies. Moreover, possible mechanisms underlying the observed U-shaped association between serum uric acid and CVD mortality in men should be elaborated.

Language and Style: Minor typographical and grammatical errors should be carefully revised throughout the manuscript. For example, in the Abstract, the sentence “Among them, 2 article was assess as ‘Very high risk of bias’, 8 ‘High risk of bias’, 2 ‘Some concerns’” requires correction for grammar and style. A comprehensive language edit is recommended.Please also consider that Numbers with four or more digits should have a comma inserted at the thousands place for readability, such as 1,234 or 10,000.

Reporting Standards and Methodological Transparency: In addition to PRISMA reporting, adherence to the MOOSE guidelines (Stroup et al., JAMA 2000) should be explicitly documented, in line with the reporting standards referenced in the registered protocol. This will strengthen the methodological transparency and ensure consistency with accepted standards for meta-analyses of observational studies.

Reviewers' comments:

Reviewer's Responses to Questions

**Comments to the Author**

1. Does the manuscript adhere to the experimental procedures and analyses described in the Registered Report Protocol?

If the manuscript reports any deviations from the planned experimental procedures and analyses, those must be reasonable and adequately justified.

Reviewer #1: Yes

Reviewer #2: Partly

Reviewer #3: Yes

2. If the manuscript reports exploratory analyses or experimental procedures not outlined in the original Registered Report Protocol, are these reasonable, justified and methodologically sound?

A Registered Report may include valid exploratory analyses not previously outlined in the Registered Report Protocol, as long as they are described as such.

Reviewer #1: Yes

Reviewer #2: Yes

Reviewer #3: Yes

3. Are the conclusions supported by the data and do they address the research question presented in the Registered Report Protocol?

The manuscript must describe a technically sound piece of scientific research with data that supports the conclusions. The conclusions must be drawn appropriately based on the research question(s) outlined in the Registered Report Protocol and on the data presented.

Reviewer #1: Yes

Reviewer #2: Yes

Reviewer #3: Yes

4. Have the authors made all data underlying the findings in their manuscript fully available?

Reviewer #1: Yes

Reviewer #2: Yes

Reviewer #3: Yes

5. Is the manuscript presented in an intelligible fashion and written in standard English?

Reviewer #1: Yes

Reviewer #2: No

Reviewer #3: Yes

6. Review Comments to the Author

Please use the space provided to explain your answers to the questions above. (Please upload your review as an attachment if it exceeds 20,000 characters)

Reviewer #1: The present is an intetesting paper aiming to evalaute impact of uricemia on CV outcomes. It is sound from a methodological point of view.

Some issues should be addressed

1) metaregression analysis for age, gender and BMI should be performed

2) more data about baseline features of each study should be added

3) recently bempedoic acid has emerged as a relevant therapy to reduce cholesterol levels, although increasing that of uric acid. Authors should comment on this (quote and comment on PMID: 38017541)

Reviewer #2: This manuscript addresses an important clinical and epidemiological question regarding the association between hyperuricemia and cardiovascular outcomes. The authors conducted a systematic review and dose-response meta-analysis with registration in PROSPERO and application of ROBINS-E and GRADE, which is commendable. The study is generally well-structured and the analyses are extensive, including sex-specific and dose-response models.

However, I suggest to adress the following points before this work can be considered for publication.

1. While the outcomes are indeed defined in both the main manuscript and the supplementary material, there is considerable heterogeneity in how individual studies defined them. Some cohorts relied on ICD codes, others on clinical/ECG confirmation, and some even included procedures such as CABG or PCI as part of the endpoint. Older cohorts often applied broader clinical definitions such as “typical angina” or “vascular disease.” This variability limits the comparability of studies and likely contributes to the high heterogeneity observed in pooled analyses. The authors should more explicitly acknowledge this as a major limitation. Furthermore, according to the registered protocol, the primary outcome was CHD, whereas the manuscript expands the scope to include additional outcomes (CVD, CVD death, MI). This expansion should be transparently reported as a deviation from the original protocol.

2. Some analyses, particularly the dose–response models, report very high heterogeneity (I² >90%). While subgroup analyses by sex were conducted, this is insufficient to explain such variability. Further exploration using meta-regression (e.g., by age distribution, mean follow-up duration, study region, publication decade, and level of confounder adjustment) would strengthen the analysis and provide more insight into sources of heterogeneity.

3. The GRADE assessments indicate that the overall certainty of evidence is low to very low. It would be advisable to include a stronger disclaimer regarding the potential for residual confounding and the inherent limitations of observational study designs.

4. The Supplementary Table S1 shows wide variability in cut-offs (5.2–7.7 mg/dL, sex-specific thresholds, SD-based definitions). This heterogeneity undermines comparability and external validity. The included studies also used diverse uric acid measurement methods (enzymatic, phosphotungstic, colorimetric, or unreported). This introduces potential misclassification bias. The impact of assay heterogeneity could be discussed.

5. The term hyperuricemia is introduced early in the Introduction, but its definition (“elevated serum uric acid levels, a condition known as hyperuricemia”) only appears in the third paragraph. For clarity and better framing, the definition should be provided the first time the term appears, ideally in the opening paragraph, to ensure readers understand the concept from the outset.

6. Several sentences are long and complex, which makes the manuscript difficult to read. Simplification and editing for concise scientific English are recommended. Topographical and grammatical errors should be corrected.

7. Abbreviations should be used consistently (e.g., CHD, CVD, MI).

8. The authors are encouraged to ensure that all figures (including supplementary figures) are cited in the text in numerical order at the point of first mention. This will improve clarity and consistency for the reader.

9. Supplementary Table S1 is dense and difficult to interpret. Organizing it by outcome type (CHD, CVD, MI) or by decade/region would improve readability.

10. Figures S1–S30 present multiple subgroup and sensitivity analyses, but the number of studies (k) and participants (n) contributing to each analysis is not always clear. This information should be explicitly included in each figure legend. In addition, not all supplementary figures are cited in the manuscript text. For clarity and transparency, each figure should be referenced at least once, in sequential order, at the relevant point in the Results or Discussion.

Reviewer #3: This study is understood to be a systematic review and meta-analysis that examines the relationship between hyperuricemia and cardiovascular disease risk.

The research procedures, methodologies, interpretation of results, and discussion were conducted with great fidelity, including prior protocol registration, making this an exceptionally well-executed and high-quality study. A notable strength of the text is its comprehensive inclusion of recent research findings, which serves to enhance its credibility and relevance.

While the paper meets the standards for publication in an international journal at present, the following suggestions are offered to enhance its value further.

Minor Point

A distinctive discovery from this study was the U-shaped association observed between serum uric acid levels and cardiovascular disease (CVD) mortality risk in men. In the conclusions, a discussion would be beneficial regarding physiologically plausible mechanisms that could explain why low uric acid levels are associated with an increased risk of cardiovascular disease (CVD) mortality.

7. PLOS authors have the option to publish the peer review history of their article (what does this mean? ). If published, this will include your full peer review and any attached files.

**Do you want your identity to be public for this peer review?** For information about this choice, including consent withdrawal, please see our Privacy Policy .

Reviewer #1: **Yes: ** fabrizio d'ascenzo

Reviewer #2: No

Reviewer #3: No

---

## [Author Response · Author response to Decision Letter 1]

14 Oct 2025

Additional Editor Comments :

In this manuscript, Lyu et al. present a systematic review and meta-analysis of observational studies evaluating the association between hyperuricemia and coronary heart disease (CHD). By systematically searching Medline, the Cochrane Library, Embase, and two trial registration databases from inception to June 30, 2025, the authors identified 42 articles corresponding to 39 unique studies, encompassing a total of 1,082,880 participants. The pooled results demonstrated a statistically significant association between hyperuricemia and increased risk of CHD, as well as its individual components (CHD death, cardiovascular death, myocardial infarction). The dose–response analyses further suggested a linear relationship between serum uric acid (UA) and cardiovascular disease (CVD) risk, and a U-shaped association between serum UA and risk of CVD death among men. Despite the relevance of the research question and the adherence to a registered protocol, the overall certainty of the evidence was limited by the observational study designs. Several key aspects require clarification and further development before the manuscript can be considered suitable for publication.

Response: We sincerely appreciate the editor and reviewers for their thorough and professional evaluation of our manuscript. In particular, we are grateful for the insightful guidance regarding heterogeneity, meta-regression, and the organization of the Supplementary Materials. These comments prompted us to reexamine all included studies and perform additional analyses. Although the overall conclusions remain largely unchanged, we are now more confident in their robustness and reliability. We believe that the constructive feedback and our corresponding revisions have substantially improved the quality of the manuscript. Please review.

Risk of Bias: Although the original protocol did not specify separate analyses for CHD-related outcomes, the authors expanded their work to include CHD death, CVD death, and myocardial infarction. While this approach is wisely comprehensive, it raises concerns regarding potential spin bias in relation to the primary outcome (CHD). Importantly, substantial heterogeneity was observed across included studies, arising from differences in association measures, uric acid assessment methods, and other study-level factors. The sources of this statistical heterogeneity should be explicitly examined and reported in relation to study characteristics and association measurements (HR,OR,RR). Furthermore, given the relevance of confounding factors in the relationship between hyperuricemia and CHD, meta-regression analyses should be performed at least for age, sex, and body mass index (BMI).

Response: Thanks for your comment. We have noticed this from the comments of reviewer 1 and reviewer 2. We have carefully reviewed all included studies, extracted necessary data, and conducted additional analyses. As we found, there was only 1 article involved the odds ratio (OR) to report their finding (Shiozaki 2013, published in Japanese), 6 involved the risk ratio (RR), 4 did not report, while the other 31 involved the hazard ratio (HR). The subgroup analysis of RR or HR was reported as Supplementary Material, and the heterogeneities of CHD death and CVD death from HR subgroup significantly decreased to 0%, while the other heterogeneities was still observed by assessing I2.

We also summarized data about the regions, the uric acid assessment methods, the count of adjusted factors, follow-up duration, BMI, age, and sex. We found 20 articles reported results from Asia, 13 from Europe, and 9 from North America. The descriptions on the uric acid assessment methods were not clear enough in most of the articles. With the help of Google, we found 11 articles used the uricase-peroxidase method, 6 used colorimetric method, 6 used phosphotungstic acid method, 5 used uricase method, and 14 with missing information. A total of 41 articles reported the adjusted factors in their statistical models, ranged from 0 to 24. For the follow-up duration, 26 articles reported the time range (from 4 to 27 years), 17 reported the average follow-up years (ranged from 2.8 to 16.8 years), 1 reported the median follow-up duration, and 1 with missing information. Considering the BMI data, 4 articles reported the BMI data of the event-group and nonevent-group, 3 reported categorized data but not mean with SD, 8 just reported the BMI data of the total population only, and 19 articles reported BMI data of the hyperuricemia population and reference population. A total of 35 articles reported the age of the included population, while only 1 article did not report the sex distribution. The above-mentioned information has been added to the tables of the manuscript or the Supplementary Materials. We found only 14 articles with available data of age, sex, and BMI. Therefore, we found it hard to perform satisfactory meta-regression.

We performed meta-regressions with age, sex, and BMI, for CHD, CHD death, CVD, CVD death, and MI, for the pooled estimations comparing participants with hyperuricemia with controls. Among the 5 outcomes, only the data of CHD death and MI were available, and found that higher proportion of male might increase the risk of CHD death among participants with hyperuricemia (β = -0.91, 95%CI -1.66 to -0.15, p = 0.018). No significant association was found for MI. When performing meta-regression without BMI, we found tiny increasing risk of CVD among male participants with hyperuricemia (β = 0.00, 95%CI -0.01 to 0.00, p = 0.043). We also performed meta-regression to test the potential confounding factors on the association between per unit increase of uric acid and the risk of cardiovascular outcome. For all 5 outcomes, BMI data was not enough to perform meta-regression, while no significant association was found on age or sex.

In conclusion, we have tried our best to explore the possible source of heterogeneity. However, through additional subgroup analyses and meta-regressions, we obtained only limited findings, which did not sufficiently explain the majority of the observed heterogeneity. We have added the above findings to the manuscript or the Supplementary Material, and added text in the Discussion section to clearly describe and report our findings to the readers. I hope this would be helpful for the further investigators. Please review.

Discussion: The Discussion would benefit from additional depth. Specifically, the clinical relevance of the findings should be contextualized in light of potential therapeutic strategies. Moreover, possible mechanisms underlying the observed U-shaped association between serum uric acid and CVD mortality in men should be elaborated.

Response: Thanks for your comment. We have searched the PubMed for articles on potential therapeutic strategies. We found that most of the articles in this field involved participants with cardiovascular disease and hyperuricemia or gout (has been discussed in our Discussion section), but not population with hyperuricemia and free of cardiovascular disease. Thus, we added the following text: “Currently, there is a substantial body of research on urate-lowering therapy for gout[91], but studies specifically addressing the management of asymptomatic hyperuricemia remain limited. Our study reports that hyperuricemia is associated with an increased risk of cardiovascular outcomes, yet it remains unclear whether urate-lowering interventions in this high-risk population would confer cardiovascular benefits, and which urate-lowering therapies would be better. We encourage researchers to conduct large-scale, long-term cohort studies in this field to evaluate the urate-lowering effects of various pharmacological and non-pharmacological interventions and to assess their long-term impact on cardiovascular outcomes in high-risk populations, thereby providing evidence for future clinical practice.”

We carefully reviewed the literature and attempted to interpret the finding of U-shaped association. However, as presented in our newly added Supplementary Figure S4, a similar U-shaped association was also observed for CHD mortality in men (p>0.05), whereas the risk of CHD itself exhibited an inverted U-shaped association (p>0.05). These patterns were not observed among women. Although we explored potential explanations for the U-shaped association, we were unable to further clarify the reasons for the inverted U-shaped relationship for CHD risk in men, or why a similar pattern was not observed in women. We acknowledge this limitation in the Discussion section and emphasize that our findings are based on limited data, highlighting the need for further studies to validate these results. For this reason, this finding was not emphasized in the Conclusion section or the Abstract section. We also anticipate that future mechanistic investigations will provide a better understanding of the underlying causes of the U-shaped association. We have added the following text in the Discussion section: “Our study identified a U-shaped association between serum UA levels and the risk of CVD mortality among men. As shown in Supplementary Figure S4, a similar U-shaped pattern was also observed for CHD mortality in men (p>0.05), whereas the risk of CHD itself demonstrated an inverted U-shaped association (p>0.05). Interestingly, these associations were not observed among women. Similar pattern has been observed from other studies as well, including kidney failure and mortality in chronic kidney disease[73], composite outcome of COVID-19 patients[74], etc. The relationship between hyperuricemia and endothelial dysfunction or vascular injury has been well recognized[75]. Some studies have suggested that uric acid possesses physiological antioxidant[76] and immunomodulatory properties[77], and that excessively low UA levels may impair free radical scavenging capacity and reduce immune regulation, ultimately leading to endothelial damage. This mechanism might explain the increased CVD mortality observed in men with low uric acid levels. However, it does not account for the inverted U-shaped association with CHD risk in men, nor the absence of similar patterns in women. It should be acknowledged that the number of original studies included in the dose-response meta-analysis was limited, which may affect the stability of the results. Further studies with high-quality are warranted to verify or correct our findings. We also anticipate that future research on uric acid will provide a more comprehensive understanding of its dual effects at different concentration levels.” Please review.

Language and Style: Minor typographical and grammatical errors should be carefully revised throughout the manuscript. For example, in the Abstract, the sentence “Among them, 2 article was assess as ‘Very high risk of bias’, 8 ‘High risk of bias’, 2 ‘Some concerns’” requires correction for grammar and style. A comprehensive language edit is recommended.Please also consider that Numbers with four or more digits should have a comma inserted at the thousands place for readability, such as 1,234 or 10,000.

Response: Thanks for the comment on language and style. We have carefully revised the whole text accordingly. The commas have been inserted. We have also invited a researcher (Yan Li from the Yong Loo Lin School of Medicine, National University of Singapore) to help us revise the manuscript. We hope the quality of language has been improved with her help. Please review.

Reporting Standards and Methodological Transparency: In addition to PRISMA reporting, adherence to the MOOSE guidelines (Stroup et al., JAMA 2000) should be explicitly documented, in line with the reporting standards referenced in the registered protocol. This will strengthen the methodological transparency and ensure consistency with accepted standards for meta-analyses of observational studies.

Response: Thanks for the comment. We have actually conducted this systematic review according to the MOOSE guidelines, but forgot to mention it in the manuscript. We are really sorry for this mistake. We have revised the text in the first paragraph of the Methods section as “……and the Meta-analysis Of Observational Studies in Epidemiology (MOOSE) guidelines [27] ……”, and uploaded the MOOSE checklist PDF file (MOOSE checklist.pdf) as attachment file. Please review.

Review Comments to the Author

Please use the space provided to explain your answers to the questions above. (Please upload your review as an attachment if it exceeds 20,000 characters)

Reviewer #1: The present is an intetesting paper aiming to evalaute impact of uricemia on CV outcomes. It is sound from a methodological point of view.

Response: Thanks for your valuable comments on our manuscript. We appreciate your patience on our work. We have carefully revised our manuscript and performed additional analyses following your guidance. We believe that the quality of the manuscript has been improved after this revision. Please review.

Some issues should be addressed

1) metaregression analysis for age, gender and BMI should be performed

Response: Thanks for your comment. Following the comments from you and the reviewer 2, we extracted more information from the articles to perform subgroup analyses and meta-regressions.

We have summarized data about the regions, the uric acid assessment methods, the count of adjusted factors, follow-up duration, BMI, age, and sex. We found 20 articles reported results from Asia, 13 from Europe, and 9 from North America. The descriptions on the uric acid assessment methods were not clear enough in most of the articles. With the help of Google, we found 11 articles used the uricase-peroxidase method, 6 used colorimetric method, 6 used phosphotungstic acid method, 5 used uricase method, and 14 with missing information. A total of 41 articles reported the adjusted factors in their statistical models, ranged from 0 to 24. For the follow-up duration, 26 articles reported the time range (from 4 to 27 years), 17 reported the average follow-up years (ranged from 2.8 to 16.8 years), 1 reported the median follow-up duration, and 1 with missing information. Considering the BMI data, 4 articles reported the BMI data of the event-group and nonevent-group, 3 reported categorized data but not mean with SD, 8 just reported the BMI data of the total population only, and 19 articles reported BMI data of the hyperuricemia population and reference population. A total of 35 articles reported the age of the included population, while only 1 article did not report the sex distribution. The above-mentioned information has been added to the tables of the manuscript or the Supplementary Materials. We found only 14 articles with available data of age, sex, and BMI. Therefore, we found it hard to perform satisfactory meta-regression, especially for BMI.

We performed meta-regressions with age, sex, and BMI, for CHD, CHD death, CVD, CVD death, and MI, for the pooled estimations comparing participants with hyperuricemia with controls. Among the 5 outcomes, only the data of CHD death and MI were available, and found that higher proportion of male might increase the risk of CHD death among participants with hyperuricemia (β = -0.91, 95%CI -1.66 to -0.15, p = 0.018). No significant association was found for MI. When performing meta-regression without BMI, we found tiny increasing risk of CVD among male participants with hyperuricemia (β = 0.00, 95%CI -0.01 to 0.00, p = 0.043). We also performed meta-regression to test the potential confounding factors on the association between per unit increase of uric acid and the risk of cardiovascular outcome. For all 5 outcomes, BMI data was not enough to perform meta-regression, while no significant association was found on age or sex.

2) more data about baseline features of each study should be added

Response: Thanks for your comment. We have established the data extraction plan referring to the previous systematic reviews in this field. Your suggestions have made us realize that the information we have summarized is insufficient. Following your suggestion, we have added as much information as possible, including the BMI data, outcome type (HR, OR, or RR), uric acid assessment methods, ori

---

## [Decision Letter · Decision Letter 1]

4 Nov 2025

Association of hyperuricemia with coronary heart disease and other cardiovascular outcomes: A systematic review and dose-response meta-analysis

PONE-D-25-42229R1

Dear Dr. Lyu,

We’re pleased to inform you that your manuscript has been judged scientifically suitable for publication and will be formally accepted for publication once it meets all outstanding technical requirements.

Kind regards,

Andres Mauricio Acevedo-Melo, M.D.

Academic Editor

PLOS ONE

Additional Editor Comments (optional):

Reviewers' comments:

Reviewer's Responses to Questions

**Comments to the Author**

1. Does the manuscript adhere to the experimental procedures and analyses described in the Registered Report Protocol?

If the manuscript reports any deviations from the planned experimental procedures and analyses, those must be reasonable and adequately justified.

Reviewer #1: Yes

Reviewer #2: Yes

Reviewer #3: Yes

2. If the manuscript reports exploratory analyses or experimental procedures not outlined in the original Registered Report Protocol, are these reasonable, justified and methodologically sound?

A Registered Report may include valid exploratory analyses not previously outlined in the Registered Report Protocol, as long as they are described as such.

Reviewer #1: Yes

Reviewer #2: Yes

Reviewer #3: Yes

3. Are the conclusions supported by the data and do they address the research question presented in the Registered Report Protocol?

The manuscript must describe a technically sound piece of scientific research with data that supports the conclusions. The conclusions must be drawn appropriately based on the research question(s) outlined in the Registered Report Protocol and on the data presented.

Reviewer #1: Yes

Reviewer #2: Yes

Reviewer #3: Yes

4. Have the authors made all data underlying the findings in their manuscript fully available?

Reviewer #1: Yes

Reviewer #2: Yes

Reviewer #3: Yes

5. Is the manuscript presented in an intelligible fashion and written in standard English?

Reviewer #1: Yes

Reviewer #2: Yes

Reviewer #3: Yes

6. Review Comments to the Author

Please use the space provided to explain your answers to the questions above. (Please upload your review as an attachment if it exceeds 20,000 characters)

Reviewer #1: authors should be congratulated for performing such a high level revisions>consequently the paper should be accpeted

Reviewer #2: I have reviewed the authors’ detailed response to the comments provided on the previous version of the manuscript, as well as the revised submission. The authors have appropriately and comprehensively addressed all the suggestions and concerns raised in the earlier review. I commend them for the quality of their revisions and the clarity of their responses. The manuscript now meets the standards for publication; however, I recommend a minor editorial review to correct a few residual grammatical or stylistic issues before final acceptance.

Reviewer #3: The description of the mapping procedures are now adequate and the innovative approach will be of interest to readers working with mapping procedures. The manuscript has been much improved and is in a nice condition now.

7. PLOS authors have the option to publish the peer review history of their article (what does this mean? ). If published, this will include your full peer review and any attached files.

**Do you want your identity to be public for this peer review?** For information about this choice, including consent withdrawal, please see our Privacy Policy .

Reviewer #1: **Yes: ** fabrizio d'ascenzo

Reviewer #2: No

Reviewer #3: No

---

## [Editor Report · Acceptance letter]

PONE-D-25-42229R1

PLOS ONE

Dear Dr. Lyu,

I'm pleased to inform you that your manuscript has been deemed suitable for publication in PLOS ONE. Congratulations! Your manuscript is now being handed over to our production team.

Kind regards,

on behalf of

Dr. Andres Mauricio Acevedo-Melo

Academic Editor

PLOS ONE